# Predicting human decision-making across task conditions via individuality transfer

**Hiroshi Higashi*†**

The University of Osaka, Osaka, Japan

## eLife Assessment

This revised paper provides a **valuable** and novel neural network-based framework for parameterizing individual differences and predicting individual decision-making across task conditions. The methods and analyses are **solid** yet could benefit from further validation of the superiority of the proposed framework against other baseline models. With these concerns addressed, this study would offer a proof-of-concept neural network approach to scientists working on the generalization of cognitive skills across contexts.

**\*For correspondence:**
higashi@comm.eng.osaka-u.ac.jp

**Present address:** †Graduate School of Engineering, The University of Osaka, Suita, Japan

**Competing interest:** The author declares that no competing interests exist.

**Abstract** Predicting an individual's behavior in one task condition based on their behavior in a different condition is a key challenge in modeling individual decision-making tendencies. We propose a novel framework that addresses this challenge by leveraging neural networks and introducing a concept we term the 'individual latent representation'. This representation, extracted from behavior in a 'source' task condition via an encoder network, captures an individual's unique decision-making tendencies. A decoder network then utilizes this representation to generate the weights of a task-specific neural network (a 'task solver'), which predicts the individual's behavior in a 'target' task condition. We demonstrate the effectiveness of our approach in two distinct decision-making tasks: a value-guided task and a perceptual task. Our framework offers a robust and generalizable approach for parameterizing individual variability, providing a promising pathway toward computational modeling at the individual level—replicating individuals in silico.

## Introduction

Humans (and other animals) exhibit substantial commonalities in their decision-making processes. However, considerable variability is also frequently observed in how individuals perform perceptual and cognitive decision-making tasks (*Carroll and Maxwell, 1979*; *Boogert et al., 2018*). This variability arises from differences in underlying cognitive mechanisms. For example, individuals may vary in their ability or tendency to retain past experiences (*Duncan and Shohamy, 2016*; *Collins and Frank, 2012*), respond to events with both speed and accuracy (*Wagenmakers and Brown, 2007*; *Spoerer et al., 2020*), or explore novel actions (*Frank et al., 2009*). If these factors can be meaningfully disentangled, they would enable a concise characterization of individual decision-making processes, yielding a low-dimensional, parameterized representation of individuality. Such a representation could, in turn, be leveraged to predict future behaviors at an individual level. Shifting from population-level predictions to an individual-based approach would mark a significant advancement in domains where precise behavior prediction is essential, such as social and cognitive sciences. Beyond prediction, this approach offers a framework for parameterizing and clustering individuals, thereby facilitating the visualization of behavioral heterogeneity, which has applications in psychiatric analysis (*Pedersen et al., 2017*; *Dezfouli et al., 2019a*). Furthermore, this parameterization offers a

promising pathway toward computational modeling at the individual level—replicating the cognitive and functional characteristics of individuals in silico (*Shengli, 2021*).

Cognitive modeling is a standard approach for reproducing and predicting human behavior (*Navarro et al., 2006*; *Busemeyer and Stout, 2002*; *Yechiam et al., 2005*), often implemented within a reinforcement learning framework (e.g. *O'Doherty et al., 2007*; *Daw et al., 2011*; *Wilson and Collins, 2019*). However, because these cognitive models are manually designed by researchers, their ability to accurately fit behavioral data may be limited (*Fintz et al., 2022*; *Song et al., 2021*; *Miller et al., 2023*; *Eckstein et al., 2022*). A data-driven approach using artificial neural networks (ANNs) offers an alternative (*Dezfouli et al., 2019b*; *Radev et al., 2022*; *Schaeffer et al., 2020*). Unlike cognitive models, which rely on predefined behavioral assumptions (*Rmus et al., 2024*), ANNs require minimal prior assumptions and can learn complex patterns directly from data. For instance, convolutional neural networks (CNNs) have successfully replicated human choices and reaction times in various visual tasks (*Kriegeskorte, 2015*; *Rajalingham et al., 2018*; *Fel et al., 2022*). Similarly, recurrent neural networks (RNNs; *Siegelmann and Sontag, 1995*; *Cho et al., 2014*) have been applied to model value-guided decision-making tasks such as the multi-armed bandit problem (*Yang et al., 2019*; *Dezfouli et al., 2019a*). A promising approach to capturing individual decision-making tendencies while preserving behavioral consistency is to tune ANN weights using a parameterized representation of individuality.

This idea was first proposed by *Dezfouli et al., 2019a*, who employed an RNN to solve a two-armed bandit task. Their study utilized an autoencoder framework (*Rumelhart and McClelland, 1987*; *Tolstikhin et al., 2017*), in which behavioral recordings from a single session of the bandit task, performed by an individual, were fed into an encoder. The encoder produced a low-dimensional vector, interpreted as a latent representation of the individual. Similar to hypernetworks (*Ha et al., 2016*; *Karaletsos et al., 2018*), a decoder then took this low-dimensional vector as input and generated the weights of the RNN. This framework successfully reproduced behavioral recordings from other sessions of the same bandit task while preserving individual characteristics. However, since this individuality transfer has only been validated within the bandit task, it remains unclear whether the extracted latent representation captures an individual's intrinsic tendencies across a variety of task conditions.

To address this question, we aim to make the low-dimensional representation—referred to as the *individual latent representation*—robust to variations across individuals and task conditions, thereby enhancing its generalizability. Specifically, we propose a framework that predicts an individual's behaviors, not only in the same condition but also in similar yet distinct task conditions and environments. If the individual latent representation serves as a low-dimensional representation of an individual's decision-making process, then extracting it from one condition could facilitate the prediction of that individual's behaviors in another.

In this study, we define the problem of *individuality transfer across task conditions* as follows (also illustrated in *Figure 1*). We assume access to a behavioral dataset from multiple individuals performing two task conditions: a *source task condition* and a *target task condition*. We train an encoder that takes behavioral data from the source task condition as input and outputs an individual latent representation. This representation is then fed into a decoder, which generates the weights of an ANN, referred to as a *task solver*, that reproduces behaviors in the target task condition. For testing, a new individual provides behavioral data from the source task condition, allowing us to infer his/her individual latent representation. Using this representation, a task solver is constructed to predict how the test individual will behave in the target task condition. Importantly, this prediction does not require any behavioral data from the test individual performing the target task condition. We refer to this framework as *EIDT*, an acronym for encoder, individual latent representation, decoder, and task solver.

We evaluated whether the proposed EIDT framework can effectively transfer individuality in both value-guided sequential decision-making tasks and perceptual decision-making tasks. To assess its generalizability across individuals, meaning its ability to predict the behavior of previously unseen individuals, we tested the framework using a test participant pool that was not included in the dataset used for model training. To determine how well our framework captures each individual's unique behavioral patterns, we compared the prediction performance of a task solver specifically designed for a given individual with the performance of task solvers designed for other individuals. Our results

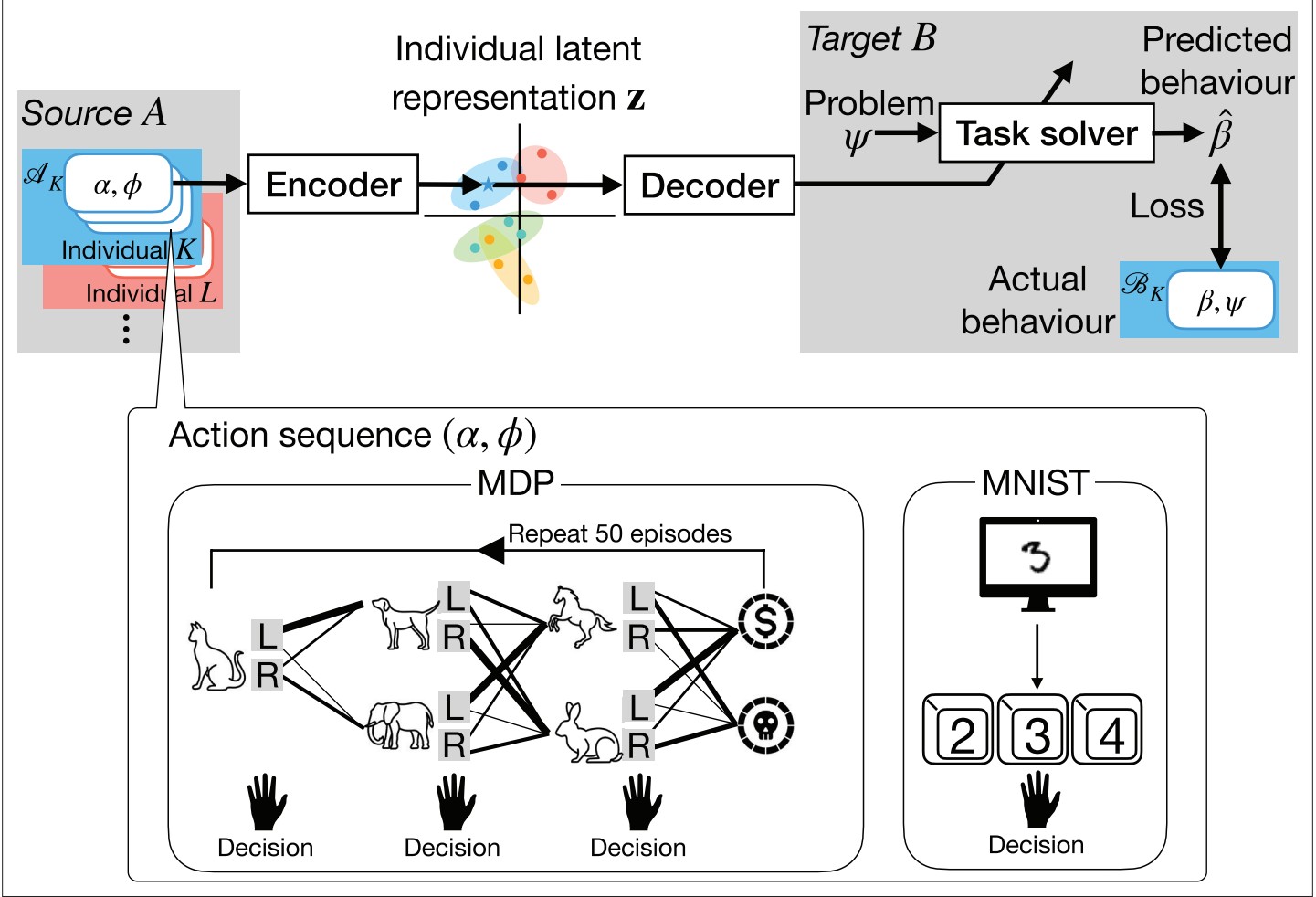

**Figure 1.** The EIDT (encoder, individual latent representation, decoder, and task solver) framework for individuality transfer across task conditions. The encoder maps action(s) $\alpha$, provided by an individual $K$ performing a specific problem $\phi$ in the source task condition $A$, into an individual latent representation (represented as a point in the two-dimensional space in the center). The individual latent representation is then fed into the decoder, which generates the weights for a task solver. The task solver predicts the behavior of the same individual $K$ in the target task condition $B$. During the training, a loss function evaluates the discrepancy between the predicted behavior $\hat{\beta}$ and the actual recorded behavior $\beta$ of individual $K$. The encoder's input is referred to as an *action sequence*, the form of which depends on task. For example, in a sequential Markov decision process (MDP) task, an action sequence consists of an environment (state transition probabilities) and a sequence of actions over multiple episodes. For a digit recognition task, it consists of a stimulus digit image and the corresponding chosen response.

indicate that the proposed framework successfully mimics decision-making while accounting for individual differences.

## Results

We evaluated our EIDT framework using two distinct experimental paradigms: a value-guided sequential decision-making task (MDP task) and a perceptual decision-making task (MNIST task). For each paradigm, we assessed model performance in two scenarios. The first, *Within-Condition Prediction*, tested a model's ability to predict behavior within a single task condition without individuality transfer. In this scenario, a model was trained on data from a pool of participants to predict the behavior of a held-out individual in that same condition. The second, *Cross-Condition Transfer*, tested the core hypothesis of individuality transfer. Here, a model used behavioral data from a participant in 'source' condition to predict that same participant's behavior in a different 'target' condition.

The prediction performance was evaluated using two metrics: the negative log-likelihood on a trial-by-trial basis, and the rate for behavior matched. The negative log-likelihood is based on the

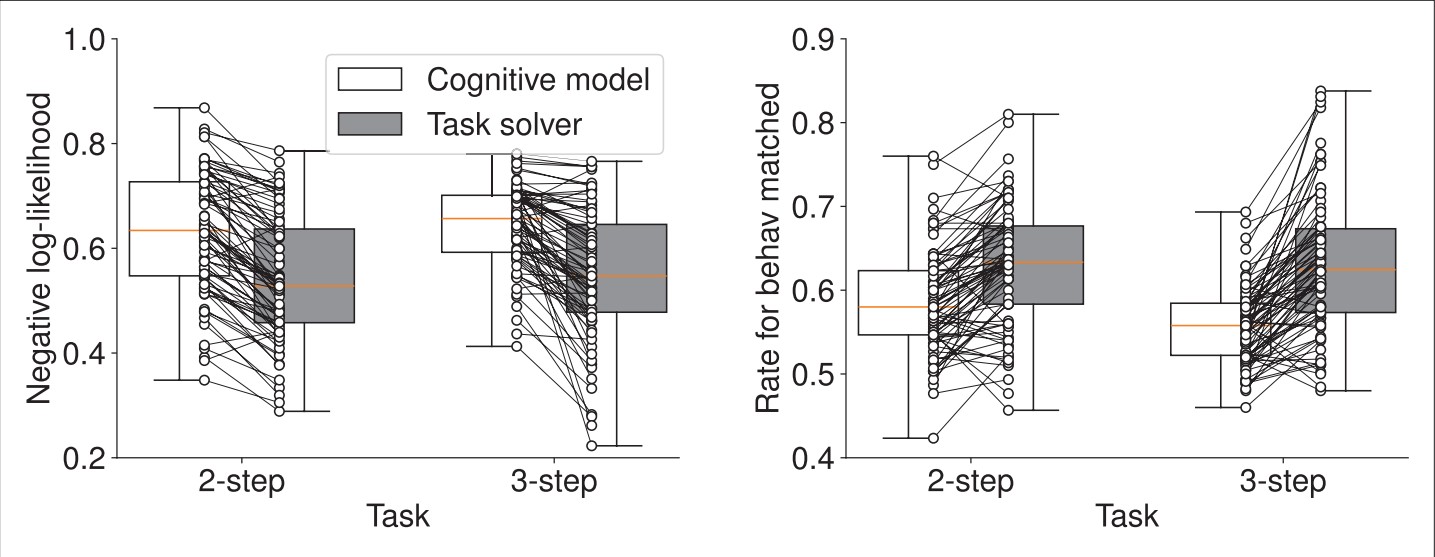

**Figure 2.** Comparison of prediction performance in Within-Condition Prediction for the MDP task. The plots show the negative log-likelihood (left) and the rate for behavior matched (right) for the average-participant cognitive model and the task solver for 2-step and 3-step conditions. Box plots indicate the median and interquartile range. Whiskers extend to the minimum and maximum values. Each connected pair of dots represents a single participant's data. The task solver demonstrates significantly better performance.

probability the model assigned to the specific action that the human participant actually took on that trial. The rate for behavior-matched measures the proportion of trials where the model's most likely action (deterministically predicted by sampling from the output probabilities) matched the participant's actual choice.

## Markov decision process (MDP) task

The dataset consisted of behavioral data from 81 participants who performed both 2-step and 3-step MDP tasks. Each participant completed three blocks of 50 episodes for each condition, resulting in 486 action sequences in total. All analyses were performed using a leave-one-participant-out cross-validation procedure. For each fold, the model was trained on 80 participants, with 90% used for training updates and 10% for validation-based early stopping.

### Task solver accurately predicts average behavior

First, we validated our core neural network architecture in Within-Condition Prediction. We trained a standard task solver, using the architecture defined in the EIDT model, on the training/validation pool ($N = 80$) to predict the behavior of the held-out participant. We compared its performance against a standard cognitive model (a Q-learning model, Cognitive model) whose parameters were averaged from fits to the same training/validation pool.

As shown in *Figure 2*, the neural network-based task solver significantly outperformed the cognitive model. A two-way (model: cognitive model/task solver, task condition: 2-step/3-step) repeated-measures (RM) ANOVA with Greenhouse-Geisser correction (significant level was 0.05) revealed a significant effect of the model on both negative log-likelihood (model: $F_{1,80} = 148.828$, $p < 0.001$, $\eta_G^2 = 0.143$, task condition: $F_{1,80} = 1.107$, $p = 0.296$, $\eta_G^2 = 0.002$, interaction: $F_{1,80} = 0.240$, $p = 0.626$, $\eta_G^2 < 0.001$) and the rate for behavior matched (model: $F_{1,80} = 110.684$, $p < 0.001$, $\eta_G^2 = 0.165$, task condition: $F_{1,80} = 3.914$, $p = 0.051$, $\eta_G^2 = 0.009$, interaction: $F_{1,80} = 19.059$, $p < 0.001$, $\eta_G^2 = 0.014$). This result confirms that our RNN-based architecture serves as a strong foundation for modeling decision-making in this task.

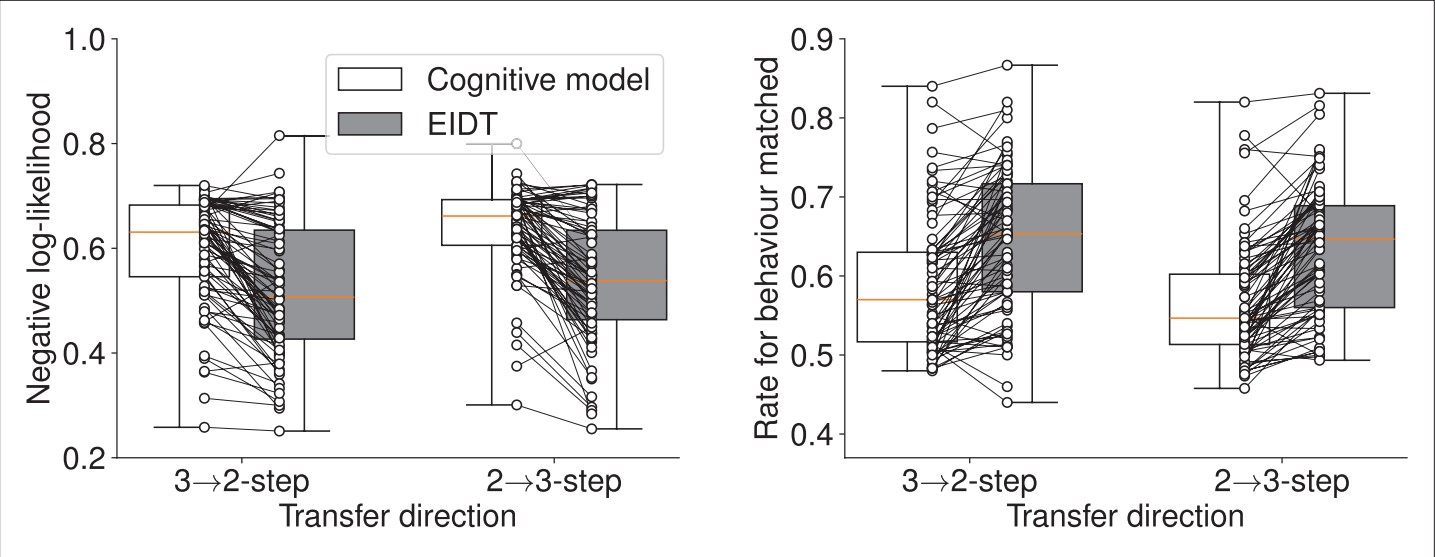

**Figure 3.** Individuality transfer performance in Cross-Condition Transfer for the MDP task. The plots compare the EIDT framework against an individualized cognitive model on negative log-likelihood (left) and rate for behavior matched (right) for both 2-step to 3-step and 3-step to 2-step transfer. Box plots indicate the median and interquartile range. Whiskers extend to the minimum and maximum values. Each connected pair of dots represents a single participant's data. The EIDT model shows superior prediction accuracy.

## EIDT enables accurate individuality transfer

Next, we tested our main hypothesis in Cross-Condition Transfer. We used the full EIDT framework to predict a participant's behavior in a target condition (e.g. 3-step MDP) using their behavioral data from a source condition (e.g. 2-step MDP). We compared the performance of two models:

### Cognitive model

A Q-learning model whose parameters ($q_{lr}$, $q_{init}$, $q_{dr}$, and $q_{it}$) were individually fitted for each participant using their data from the source condition and then applied to predict behavior in the target condition.

### EIDT

Our framework, trained on the training and validation pool using data from both source and target conditions (see *Appendix 1—figure 2*, Appendix 1 for representative training and validation curves). To predict behavior for a test participant, their individual latent representation was computed by averaging the encoder's output across all of their behavioral sequences from the source condition, and this representation was fed to the decoder to generate the task solver weights. For reference, the averaged individual latent representations are visualized in *Appendix 1—figure 3*, Appendix 1.

The EIDT framework demonstrated significantly better prediction accuracy than the individualized cognitive model (*Figure 3*). A two-way (model: cognitive model/EIDT, transfer direction: 2→3/3→2) RM ANOVA confirmed a significant effect of the model on negative log-likelihood (model: $F_{1,80} = 95.705$, $p < 0.001$, $\eta_G^2 = 0.142$, transfer direction: $F_{1,80} = 14.255$, $p < 0.001$, $\eta_G^2 = 0.019$, interaction: $F_{1,80} = 0.008$, $p = 0.012$, $\eta_G^2 = 0.002$) and the rate for behavior matched (model: $F_{1,80} = 100.843$, $p < 0.001$, $\eta_G^2 = 0.132$, transfer direction: $F_{1,80} = 13.021$, $p = 0.001$, $\eta_G^2 = 0.011$, interaction: $F_{1,80} = 0.964$, $p = 0.329$, $\eta_G^2 < 0.001$). This result indicates that EIDT successfully captures and transfers individual-specific behavioral patterns more effectively than a traditional parameter-based transfer approach.

## Latent space distance predicts transfer performance

To verify that the individual latent representation meaningfully captures individuality, we conducted a 'cross-individual' analysis. We generated a task solver using the latent representation of one participant (Participant *l*) and used it to predict the behavior of another participant (Participant *k*). We then measured the relationship between the prediction performance ($y_{k,l}$) and the Euclidean distance ($d_{k,l}$) between the latent representations of Participants *k* and *l*.

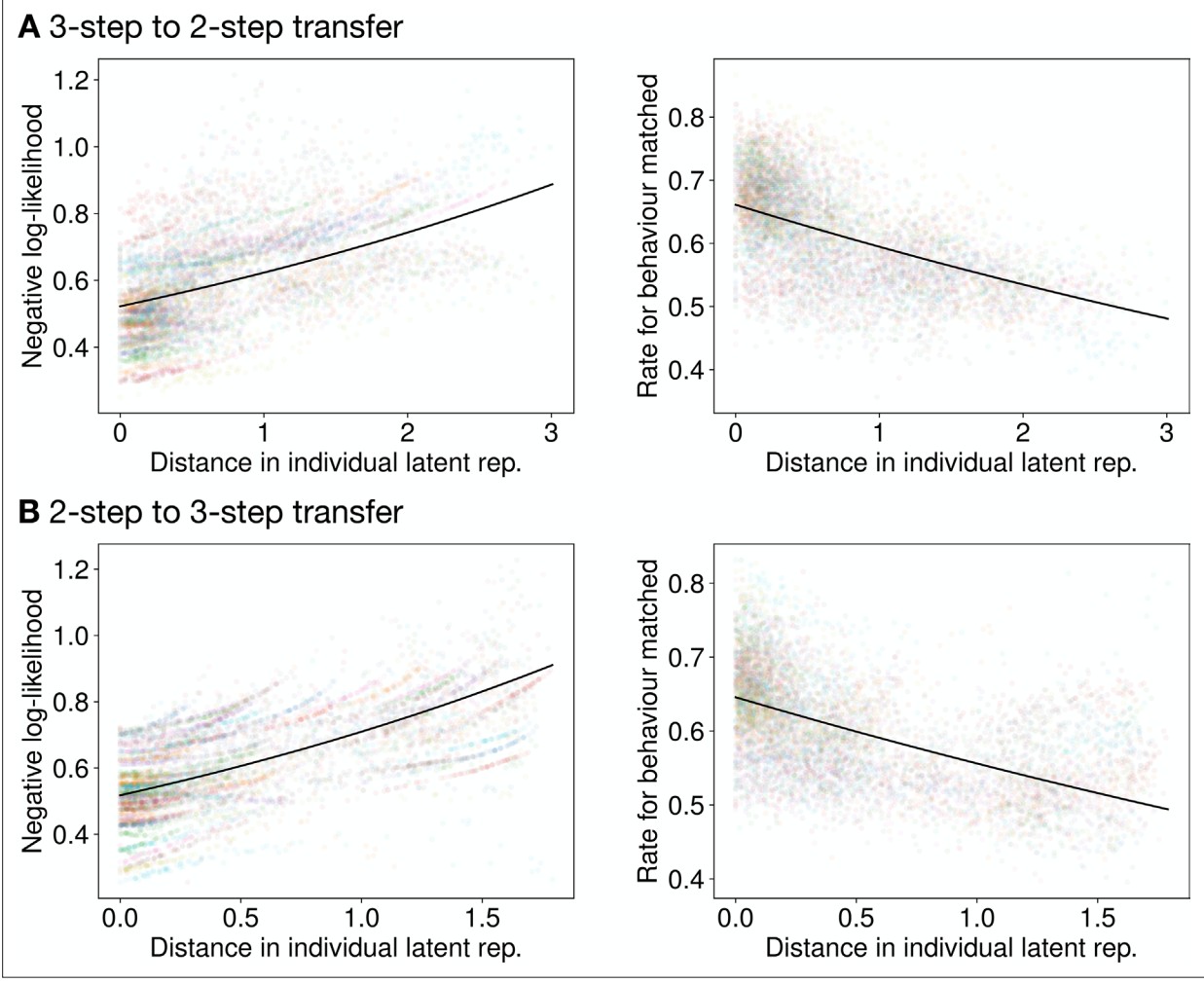

**Figure 4.** Prediction performances as functions of latent space distance in the MDP task. This cross-individual analysis shows the result of using a task solver generated from one participant to predict the behavior of another participant. The horizontal axis is the Euclidean distance between the latent representation of the two participants. The vertical axis shows the negative log-likelihood (left) and rate for behavior matched (right). Each dot represents one participant pair. Performance degrades as the distance between individuals increases, with the solid line showing the GLM fit. (**A**) 3-step to 2-step transfer. (**B**) 2-step to 3-step transfer.

As hypothesized, prediction performance was strongly dependent on this distance (**Figure 4**). We fitted the data using a generalized linear model (GLM): $y_{k,l} \sim \text{Gamma}\left(\log\left(\beta_{\text{participant}_k} + \beta_d d_{k,l} + \beta_0\right)\right)$. The fitting confirmed that distance ($d_{k,l}$) was a significant predictor: the coefficient $\beta_d$ was significantly positive for negative log-likelihood (transfer direction 3→2: $\beta_d = 0.176$, $p < 0.001$, 2→3: $\beta_d = 0.316$, $p < 0.001$) and significantly negative for the rate for behavior matched (3→2: $\beta_d = 0.106$, $p < 0.001$, 2→3: $\beta_d = -0.149$, $p < 0.001$). This indicates that prediction performance degrades as the behavioral dissimilarity (represented by distance in the latent space) between the source and target individual increases, providing direct evidence that the latent space organizes individuals by behavioral similarity.

## On-policy simulations generate human-like behavior

To assess if our model could generate realistic behavior, we conducted on-policy simulations. Task solvers specialized to each individual via EIDT performed the MDP task using the same environments as the human participants. We compared the model behavior to human behavior on two metrics: total reward per block and the rate of highly rewarding action selected in the final step.

The model-generated behaviors closely mirrored human behaviors (**Figure 5**). We found significant correlations between humans and their corresponding models in both total rewards (3→2: $R = 0.667$, $p < 0.001$; 2→3: $R = 0.593$, $p < 0.001$) and the rate of highly-rewarding action selected (3→2: $R = 0.889$,

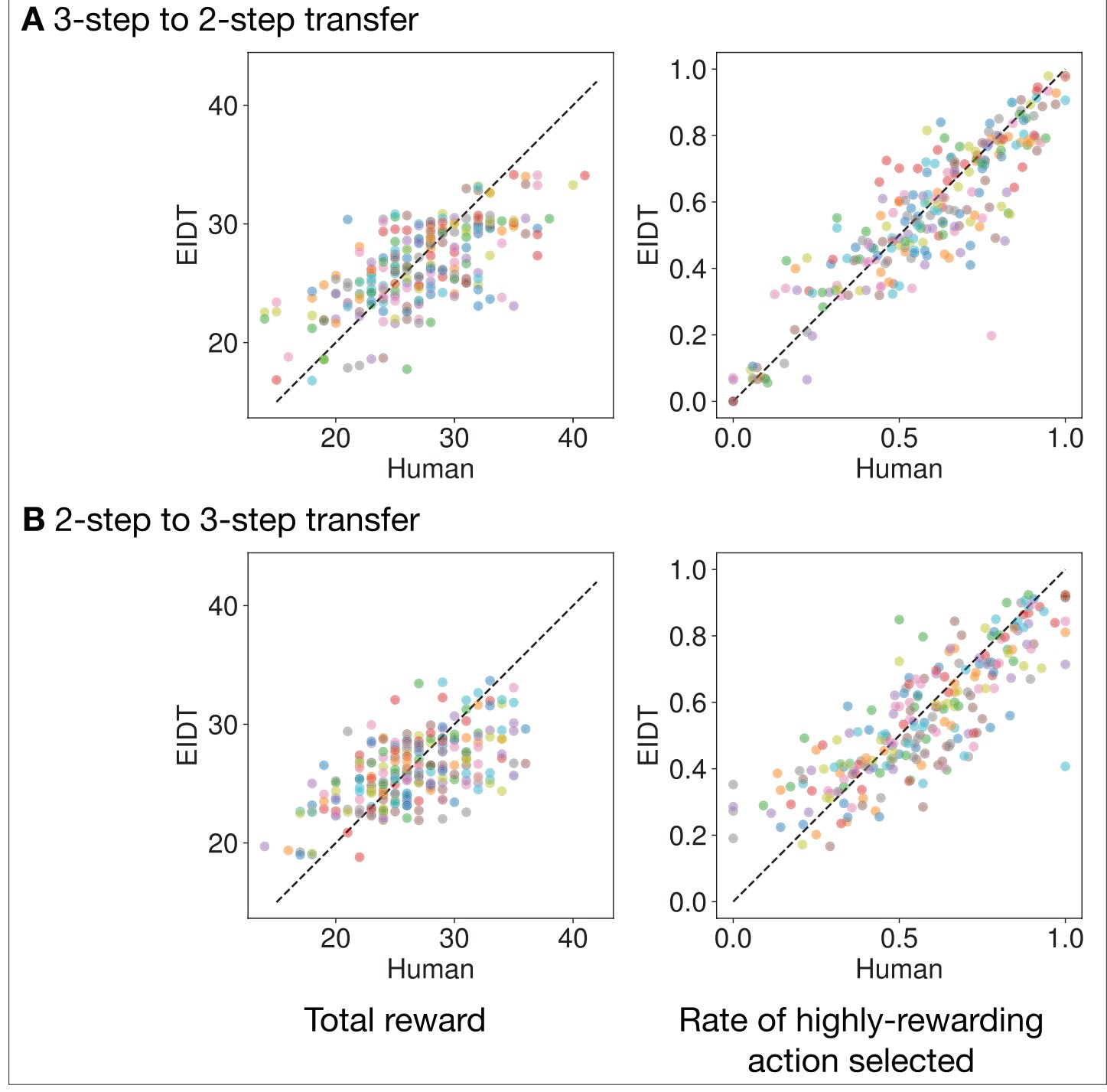

**Figure 5.** Comparison of on-policy behavior between humans and EIDT-generated task solvers. Each dot represents the performance of a single human participant (horizontal axis) versus their corresponding model (vertical axis) for one block. Plots show the total reward (left) and the rate of highly-rewarding action selected (right). (**A**) 3-step to 2-step transfer. (**B**) 2-step to 3-step transfer.

$p < 0.001$; 2→3: $R = 0.835$, $p < 0.001$). This demonstrates that the EIDT framework captures individual tendencies that generalize to active, sequential behavior generation.

### Individual latent representations reflect cognitive parameters

To better interpret the latent space, we applied our EIDT model (trained only on human data) to simulated data from 1000 Q-learning agents. The agents had known learning rates ($q_{lr}$) and inverse

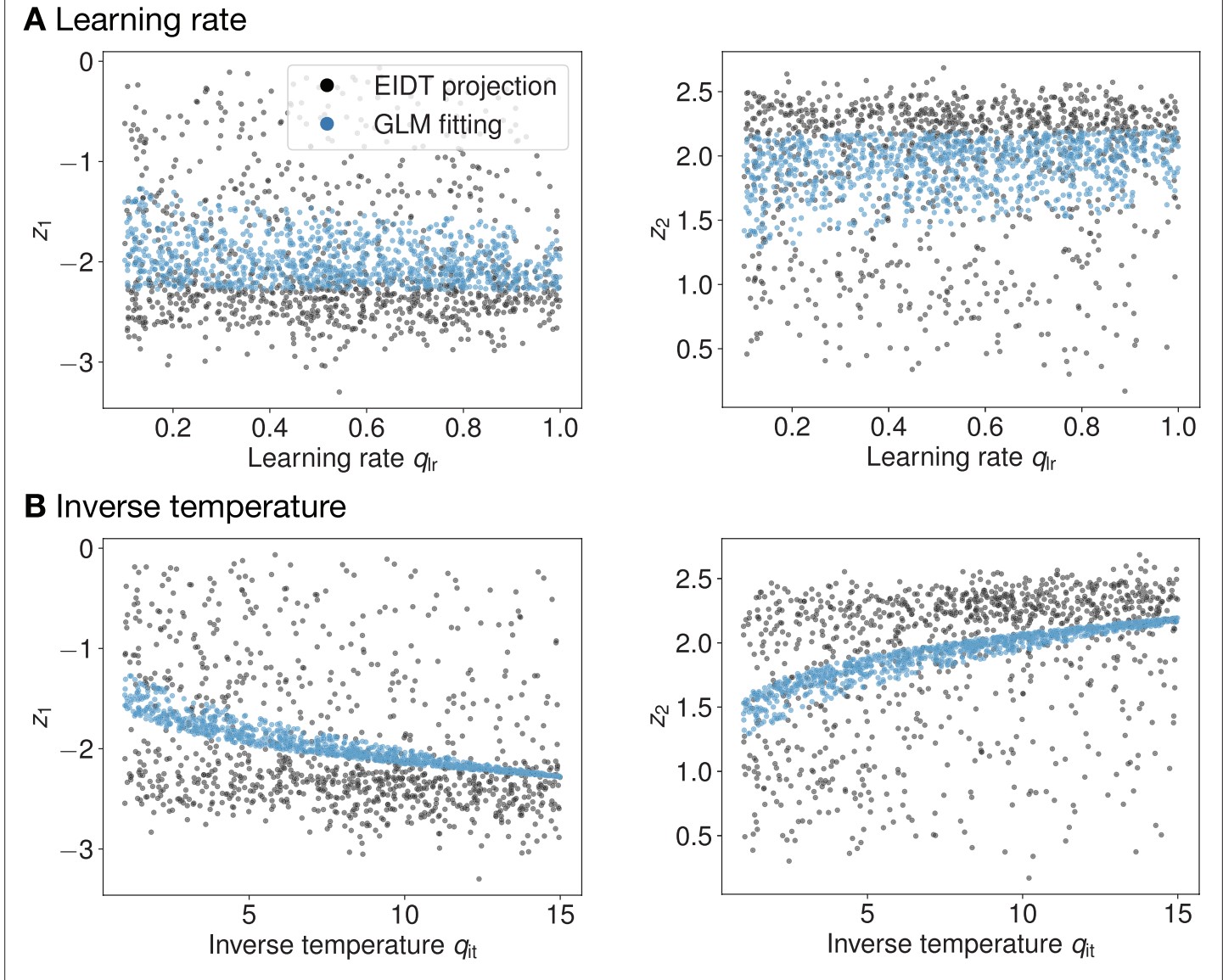

**Figure 6.** Mapping of Q-learning parameters to the individual latent space for the 3-step MDP task. Each plot shows one dimension of the latent representation ($z_1$ (left) or $z_2$ (right)) as a function of either the learning rate ($q_{lr}$, **A**) or the inverse temperature ($q_{it}$, **B**) of simulated Q-learning agents. Black dots represent the latent representation produced by the encoder from the agent's behavior. Blue dots show the fit from a GLM.

temperatures ($q_{it}$) sampled from distributions matched to human fits (*Appendix 1—figure 1*, Appendix 1). A cross-individual analysis on these agents confirmed that latent space distance predicted performance, mirroring the results from human data (*Appendix 1—figure 5*, Appendix 1).

The results revealed a systematic mapping between the cognitive parameters and the coordinates of the individual latent representation (*Figure 6* and *Appendix 1—figure 4*, Appendix 1). A GLM analysis (*Appendix 1—table 1*, Appendix 1) showed that both $q_{rl}$ and $q_{it}$ (and their interaction) were significant predictors of the latent dimensions ($z_1$ and $z_2$). This indicates that our data-driven representation captures core computational properties defined in classic reinforcement learning theory.

### Handwritten digit recognition (MNIST) task

We then sought to replicate our findings in a different domain: perceptual decision-making. We used data from *Rafiei et al., 2024*, where 60 participants identified noisy images of digits under four conditions varying in difficulty and speed-accuracy focus (EA: easy, accuracy focus, ES: easy, speed focus,

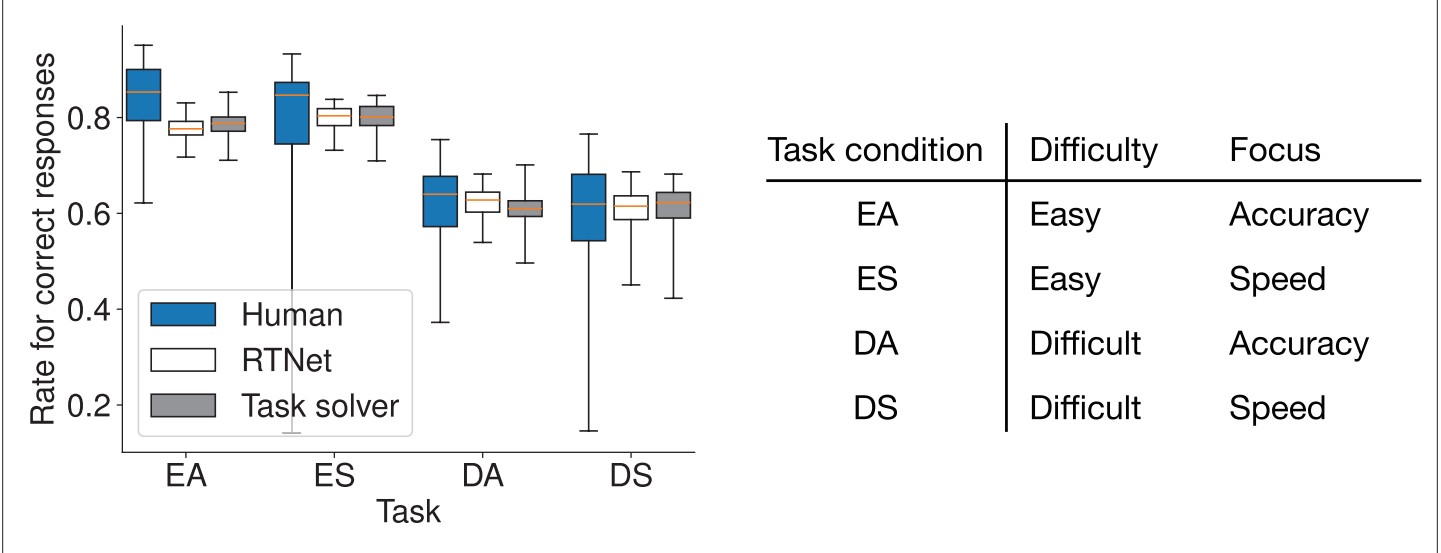

| Task condition | Difficulty | Focus |
|---|---|---|
| EA | Easy | Accuracy |
| ES | Easy | Speed |
| DA | Difficult | Accuracy |
| DS | Difficult | Speed |

**Figure 7.** Task performance (rate of correct responses) in Within-Condition Prediction for the MNIST tasks. Box plots indicate the median and interquartile range. Whiskers extend to the minimum and maximum values. Performance is compared across human participants, the RTNet model, and our task solver for the four experimental conditions (EA, ES, DA, and DS). All three show similar performance patterns.

DA: difficult, accuracy focus, and DS: difficult, speed focus). Analyses were again conducted using leave-one-participant-out cross-validation.

## Task solver outperforms RTNet

First, in Within-Condition Prediction, our base task solver demonstrated task performance (rate of correct responses indicating how accurately a human participant or model responded to the stimulus digit) comparable to human participants and established RTNet model (*Rafiei et al., 2024*; *Figure 7*). A two-way (model: human/RTNet/Task solver, task condition: EA/ES/DA/DS) RM ANOVA showed no significant effect of model type ($F_{2,118} = 1.546$, $p = 0.219$, $\eta_G^2 = 0.008$), while the task condition had a significant effect ($F_{3,177} = 866.322$, $p < 0.001$, $\eta_G^2 = 0.684$). This confirms similar task-solving ability.

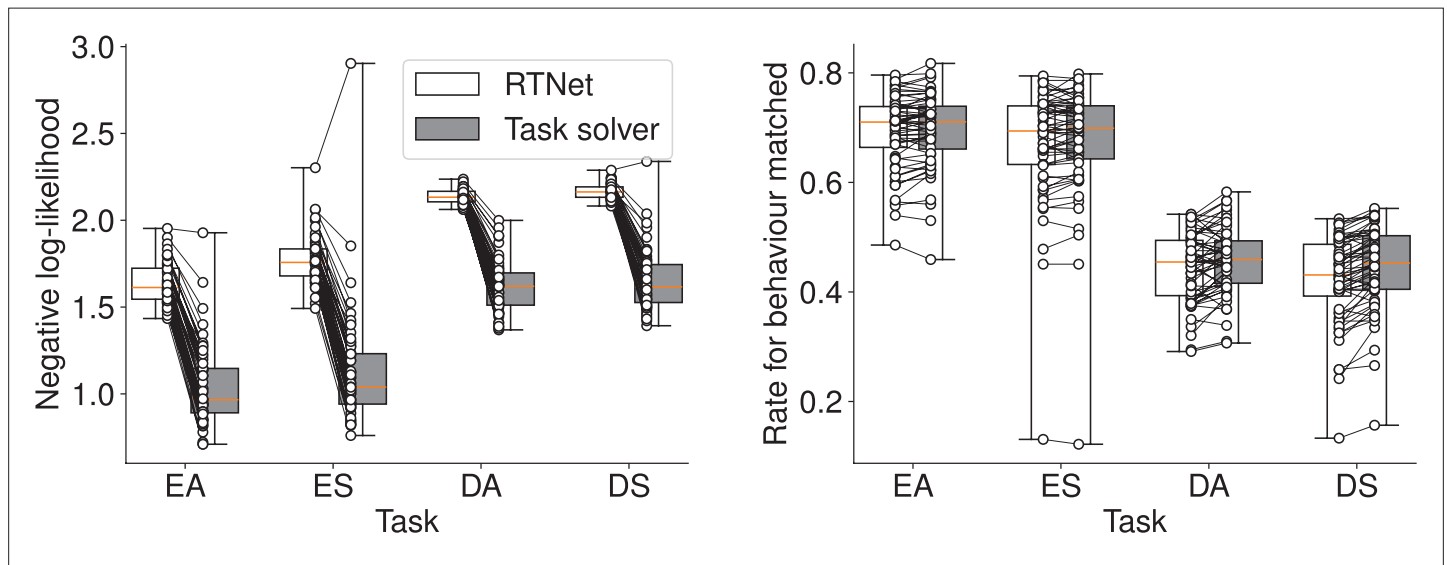

**Figure 8.** Comparison of prediction performance in Within-Condition Prediction for the MNIST task. The plots show the negative log-likelihood (left) and the rate for behavior matched (right) for the RTNet model and our task solver. Each connected pair of dots represents a single participant's data. Box plots indicate the median and interquartile range. Whiskers extend to the minimum and maximum values. The task solver achieves significantly better prediction accuracy.

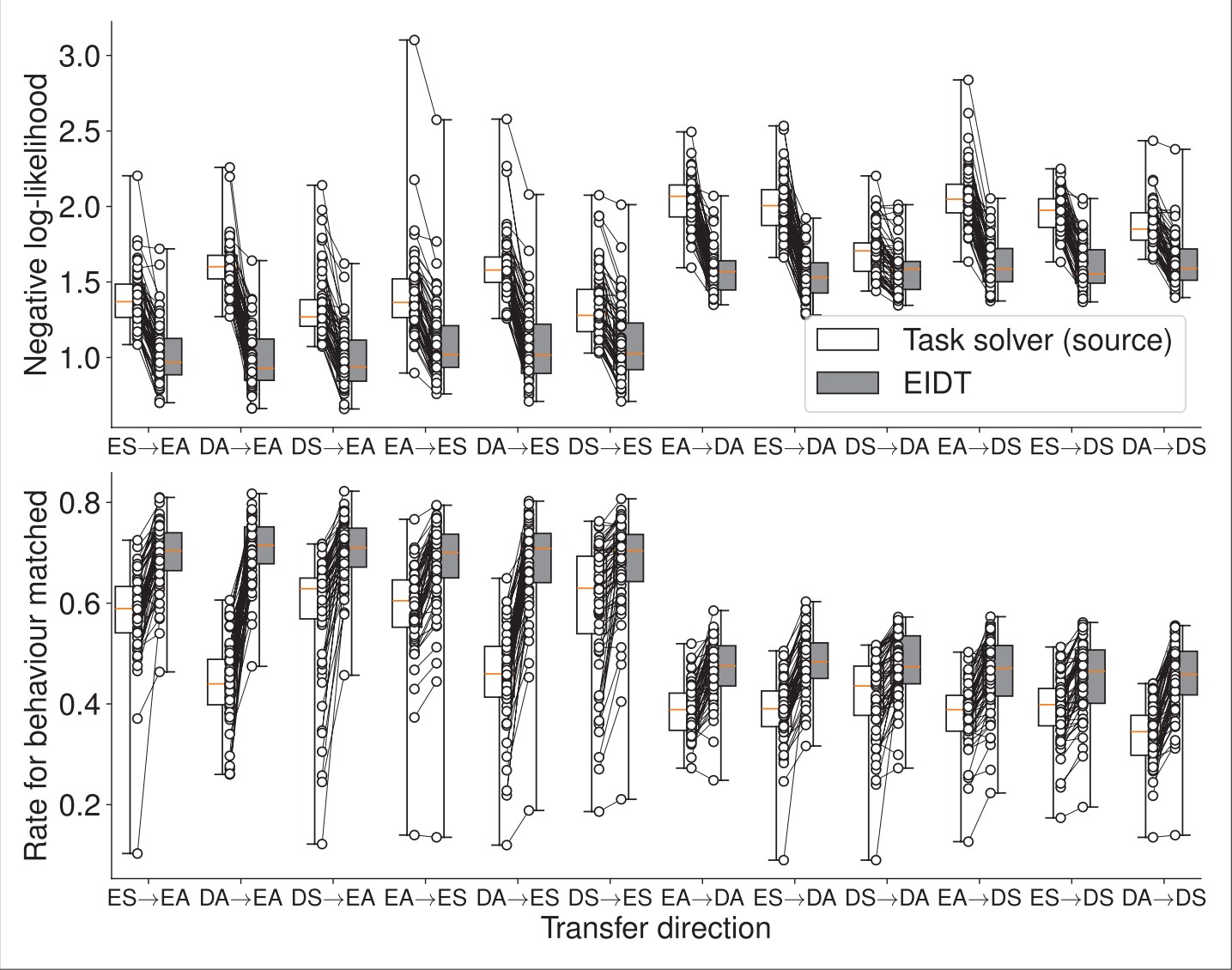

**Figure 9.** Individuality transfer performance in Cross-Condition Transfer for the MNIST task. The plots compare the EIDT framework against the task solver (source) baseline across all 12 transfer directions on negative log-likelihood (top) and rate for behavior matched (bottom). Each connected pair of dots represents a single participant's data. Box plots indicate the median and interquartile range. Whiskers extend to the minimum and maximum values. EIDT consistently demonstrates superior prediction accuracy.

However, the task solver significantly outperformed RTNet in predicting participants' trial-by-trial choices (*Figure 8*). A two-way RM ANOVA revealed significant effects on both negative log-likelihood (model: $F_{1,59} = 1312.328$, $p < 0.001$, $\eta_G^2 = 0.731$, task condition: $F_{3,177} = 460.535$, $p < 0.001$, $\eta_G^2 = 0.682$, their interaction: $F_{3,177} = 24.476$, $p < 0.001$, $\eta_G^2 = 0.026$) and the rate for behavior matched (model: $F_{1,59} = 43.544$, $p < 0.001$, $\eta_G^2 = 0.005$, task condition: $F_{3,177} = 455.728$, $p < 0.001$, $\eta_G^2 = 0.701$, their interaction: $F_{3,177} = 11.052$, $p < 0.001$, $\eta_G^2 = 0.002$). This confirms the task solver's suitability for modeling individual behavior in this task.

## EIDT accurately transfers individuality

Next, in Cross-Condition Transfer, we tested individuality transfer across all 12 pairs of experimental conditions. The full EIDT framework was compared against a baseline: a task solver (source) model trained directly on a test participant's source condition data.

The EIDT framework consistently and significantly outperformed this baseline across all transfer sets (*Figure 9*). A two-way (model: task solver/EIDT, transfer direction: 12 sets (see

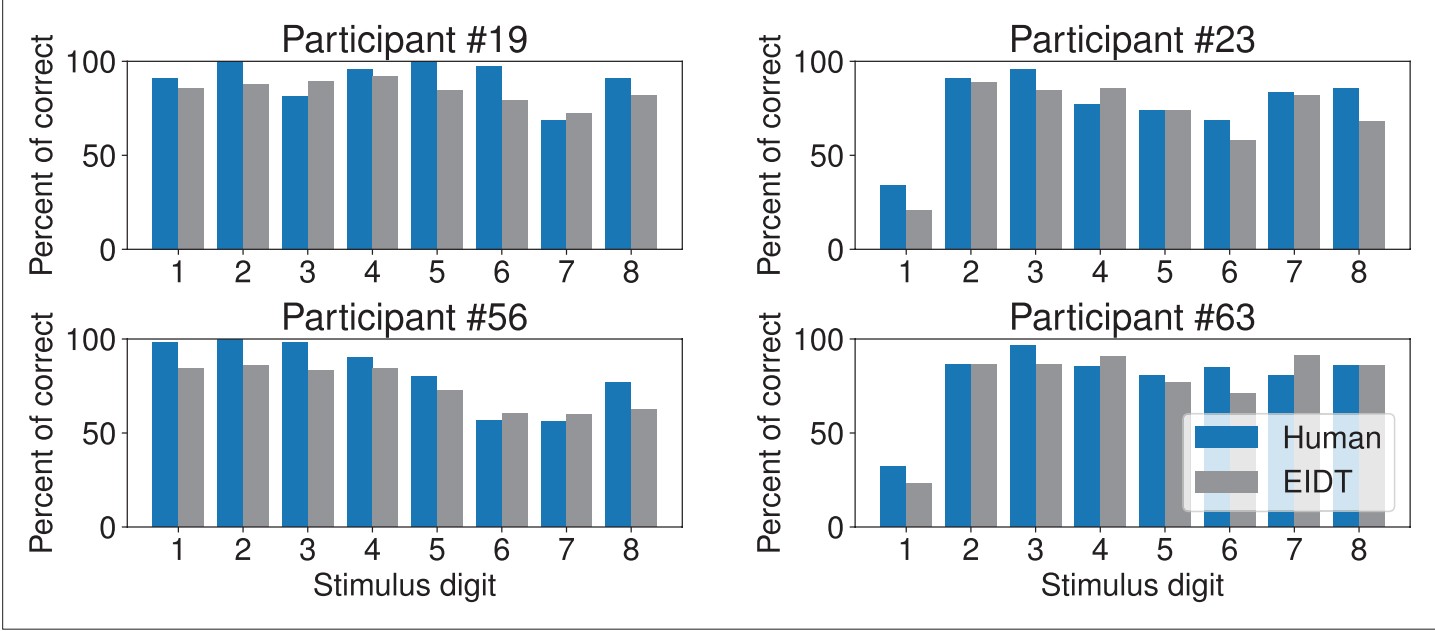

**Figure 10.** EIDT captures individual-specific error patterns in the MNIST task. The plots show the percentage of correct responses for each digit for four representative participants (blue bars) and their corresponding EIDT-generated models (gray bars). Data shown is for the ES target condition, with transfer from EA.

horizontal axis)) RM ANOVA confirmed a significant effect of the model on negative log-likelihood (model: $F_{3,177} = 2440.373$, $p < 0.001$, $\eta_G^2 = 0.800$, transfer direction: $F_{11,649} = 347.850$, $p < 0.001$, $\eta_G^2 = 0.616$, interaction: $F_{33,1947} = 336.968$, $p < 0.001$, $\eta_G^2 = 0.573$) and rate for behavior matched (model: $F_{3,177} = 2318.456$, $p < 0.001$, $\eta_G^2 = 0.798$, transfer direction: $F_{11,649} = 394.753$, $p < 0.001$, $\eta_G^2 = 0.591$, interaction: $F_{33,1947} = 355.577$, $p < 0.001$, $\eta_G^2 = 0.628$). The model was also able to reproduce idiosyncratic error patterns of individual participants, such as Participant #23's lower accuracy for digit 1 and Participant #56's difficulty with digits 6 and 7 (**Figure 10**).

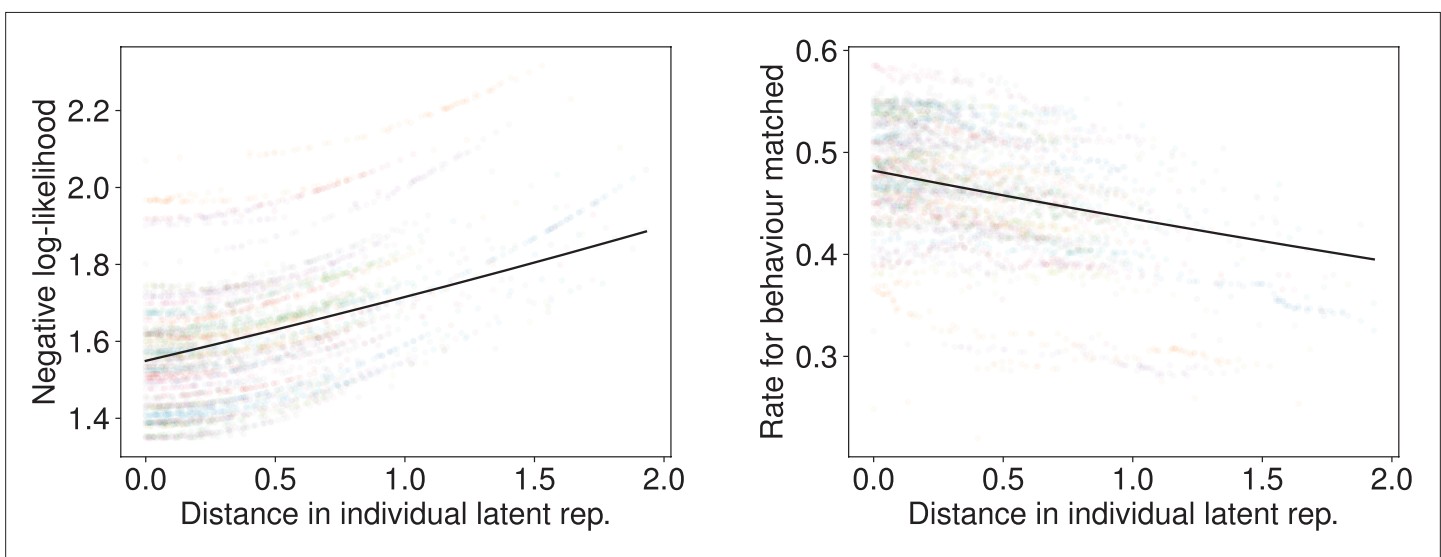

**Figure 11.** Prediction performance as a function of latent space distance in the MNIST task (transfer direction EA→DA). This cross-individual analysis shows the result of using a task solver generated from one participant to predict the behavior of another participant. The horizontal axis is the Euclidean distance between the latent representation of the two participants. The vertical axis shows the negative log-likelihood (left) and rate for behavior matched (right). Each dot represents one participant pair. Performance degrades as the distance between individuals increases, with the solid line showing the GLM fit.

#### Latent space reflects behavioral tendencies

Similar to the MDP task, a cross-individual analysis showed that the distance in the latent space was a significant predictor of prediction performance for all transfer directions (*Figure 11*; see *Appendix 1—figures 8 and 9* and *Appendix 1—table 2*, Appendix 1, for full results). This confirms that, in the perceptual domain as well, the individual latent representation captures meaningful behavioral differences that are critical for accurate prediction.

## Discussion

We proposed an EIDT framework for modeling the unique decision-making process of each individual. This framework enables the transfer of an individual latent representation from a (source) task condition to a different (target) task condition, allowing a task solver to predict behaviors in the target task condition. Several neural network techniques, such as autoencoders (*Rumelhart and McClelland, 1987*; *Tolstikhin et al., 2017*), hypernetworks (*Ha et al., 2016*), and learning-to-learn (*Wang et al., 2017*; *Song et al., 2017*), facilitate this transfer. Our experiments, conducted on both value-guided sequential and perceptual decision-making tasks, demonstrated the potential of the proposed framework in individuality transfer across task conditions.

### EIDT framework extends prior work on individuality transfer

The core concept of using an encoder-decoder architecture to capture individuality builds on the work of *Dezfouli et al., 2019a*, who applied a similar model to a bandit task. We extended this idea in three key ways. First, we validated that the framework is effective for previously unseen individuals who were not included in model training. Although these individuals provided behavioral data in the source task condition to identify their individual latent representations, their data were not used for model training. Second, we established that this transfer is effective across different experimental conditions (e.g. changes in task rules or difficulty), not just across sessions of the same task. Third, while the original work focused on value-guided tasks, we validated the framework's applicability to perceptual decision-making tasks, specifically the MNIST task. These findings establish that EIDT effectively captures individual differences across both task conditions and individuals.

### Interpreting the individual latent representation remains challenging

Although we found that Q-learning parameters were reflected in the individual latent representation, the interpretation of this representation remains an open question. Since interpretation often requires task-condition-specific considerations (*Eckstein et al., 2022*), it falls outside the primary scope of this study, whose aim is to develop a general framework for individuality transfer. Previous research (*Miller et al., 2023*; *Ger et al., 2024a*) has explored associating neural network parameters with cognitive or functional meanings. Approaches such as disentangling techniques (*Burgess et al., 2018*) and cognitive model integration (*Ger et al., 2024b*; *Tuzsus et al., 2024*; *Song et al., 2021*; *Eckstein et al., 2023*) could aid in better understanding the cognitive and functional significance of the individual latent representation.

Regarding the individual latent representation, disentanglement and separation losses (*Dezfouli et al., 2019a*) during the model training could enhance interpretability. However, we used only the reproduction loss, as defined in *Equation 5*, because interpretable parameters in cognitive models (e.g. *Daw et al., 2011*) are not necessarily independent (e.g. an individual with a high learning rate may also have a high inverse temperature *Lin et al., 2023*, resulting in these two parameters being represented with one variable).

### Why can the encoder extract individuality for unseen individuals?

Our experiments, which divided participants into training and test participant pools, demonstrated that the framework successfully extracts individuality for completely new individuals. This generalization likely relies on the fact that individuals with similar behavioral patterns result in similar individual latent representation and individuals similar to new participants exist in the training participant pool (*Yechiam et al., 2005*). This hypothesis suggests that individuals can be clustered based on behavioral patterns. Behavioral clustering has been widely discussed in relation to psychiatric conditions, medication effects, and gender-based differences (e.g. *Pedersen et al., 2017*; *van den Bos et al., 2013*;

*Sevy et al., 2007*). Our results could contribute to a deeper discussion of behavioral characteristics by clustering not only these groups but also healthy controls.

## Which processes contribute to individuality?

In the MNIST task, we assumed that individuality emerged primarily from the decision-making process (implemented by an RNN *Spoerer et al., 2020*; *Cheng et al., 2024*), rather than from the visual processing system (implemented by a CNN *Yamins and DiCarlo, 2016*). The CNN was pretrained, and the decoder did not tune its weights. Our results do not rule out the possibility that the visual system also exhibits individuality (*Koivisto et al., 2011*; *Tang et al., 2018*); however, they imply that individual differences in perceptual decision-making can be explained primarily by variations in the decision-making system (*Ratcliff and McKoon, 2008*; *Vickers, 1970*; *Yechiam et al., 2005*; *Kar et al., 2019*). This assumption provides valuable insights for research on human perception.

## Limitations

One limitation is that the source and target behaviors were performed on different conditions, but within the same task. Thus, our findings do not fully evaluate the generalizability of individuality transfer across diverse task domains. However, our framework has the potential to be applied to diverse tasks since it connects the source and target tasks via the individual latent representation and accepts completely different tasks for the source and target. A key to realizing this transfer might be ensuring that the cognitive functions, such as memory, required for solving the source and target tasks are (partially) shared. The latent representation is expected to represent individual features of these functions. Conversely, if source and target tasks require completely different functions to solve them, the transfer by EIDT would not work.

The effectiveness of individuality transfer may be influenced by dataset volume. As discussed earlier, prediction performance may depend on whether similar individuals exist in the training participant pool. In our study, 100 participants were sufficient for effective transfer. However, tasks involving greater behavioral diversity may require a substantially larger dataset.

As discussed earlier, the interpretability of the individual latent representation requires further investigation. Furthermore, the optimal dimensionality of the individual latent representation remains unclear. This likely depends on the complexity of tasks involved—specifically, the number of factors needed to represent the diversity of behavior observed in those tasks. While these factors have been explored in cognitive modeling research (e.g., *Katahira, 2015*; *Eckstein et al., 2022*), a clear understanding at the individual level is still lacking. Integrating cognitive modeling with data-driven neural network approaches (*Dezfouli et al., 2019a*; *Ger et al., 2024b*) could help identify key factors underlying individual differences in decision-making.

## Future directions

To further generalize our framework, a large-scale dataset is necessary, as discussed in the limitations. This dataset should include a large number of participants to ensure prediction performance for diverse individuals (*Peterson et al., 2021*). All participants should perform the same set of tasks, which should include a variety of tasks (*Yang et al., 2019*). Building upon our framework, where the encoder currently accepts action sequences from only a single task, a more generalizable encoder should be able to process behavioral data from multiple tasks to generate a more robust individual latent representation. To enhance the encoder, a multi-head neural network architecture (*Canizo et al., 2019*) could be utilized. An individual latent representation would enable transfer to a wider variety of tasks and allow accurate and detailed parameterization of individuals using data from only a single task.

Robust and generalizable parameterization of individuality enables computational modeling at the individual level. This approach, in turn, makes it possible to replicate individuals' cognitive and functional characteristics in silico (*Shengli, 2021*). We anticipate that it offers a promising pathway toward a new frontier: artificial intelligence endowed with individuality.

## Methods

### General framework for individuality transfer across task conditions

We formulate the problem of individuality transfer, which involves extracting an individual latent representation from a source task condition and predicting behavior in a target task condition while preserving individuality. We consider two task conditions, $A$ and $B$, which are different but related. For example, condition $A$ might be a 2-step MDP, while condition $B$ is a 3-step MDP.

The individuality transfer across task conditions is defined as follows. An individual $K$ performs a problem within condition $A$, with their behavior recorded as $\mathcal{A}_K$. Our objective is to predict $\mathcal{B}_K$, which represents $K$'s behavior when performing a task with condition $B$. To achieve this, we extract an individual latent representation $z$ from $\mathcal{A}_K$, capturing the individual's behavioral characteristics. This representation $z$ is then used to construct a task solver, enabling it to mimic $K$'s behavior in condition $B$. Since condition $A$ provides data for estimating the individual latent representation and condition $B$ is the target of behavior prediction, we refer to them as the source task condition and target task condition, respectively.

Our proposed framework for the individuality transfer consists of three modules:

**Task solver** predicts behavior in the target condition $B$.

**Encoder** extracts the individual latent representation from the source condition $A$.

**Decoder** generates the weights of the task solver based on the individual latent representation.

These modules are illustrated in *Figure 1*. We refer to this framework as EIDT, an acronym for encoder, individual latent representation, decoder, and task solver.

### Data representation

For training, we assume that behavior data from a participant pool $\mathcal{P}$ ($K \notin \mathcal{P}$), where each participant has performed both conditions $A$ and $B$. These datasets are represented as $\mathcal{A} = \{\mathcal{A}_n\}_{n \in \mathcal{P}}$ and $\mathcal{B} = \{\mathcal{B}_n\}_{n \in \mathcal{P}}$.

For each individual $n$, the set $\mathcal{A}_n$ consists of one or more sets, each containing a problem instance $\phi$ (stimuli, task settings, or environment in condition $A$) and a sequence of action(s) $\alpha$ (recorded behavioral responses). For example, in an MDP task, $\phi$ represents the Markov process (state-action-reward transition) and $\alpha$ consists of choices over multiple trials. In a simple object recognition task, $\phi$ is a visual stimulus and $\alpha$ is the participant's response to the stimulus. Similarly, $\mathcal{B}_n$ consists of a problem instance $\psi$ and an action sequence $\beta$.

### Task solver

The task solver predicts the action sequence for condition $B$ as

$$\hat{\beta} = \text{TS}(\psi; \Theta_{\text{TS}}), \tag{1}$$

where $\psi$ is a specific problem in condition $B$ and $\Theta_{\text{TS}}$ represents the solver's weights. The task solver architecture is tailored to condition $B$. For example, in an MDP task, the task solver outputs a sequence of actions in response to $\psi$. In a simple object recognition task, it produces an action based on a visual stimulus $\psi$.

### Encoder

The encoder processes an action sequence(s) $\alpha$ and generates an individual latent representation $z \in \mathbb{R}^M$ as

$$z = \text{ENC}(\alpha, \phi; \Theta_{\text{ENC}}), \tag{2}$$

where $\phi$ is a problem in condition $A$, $\Theta_{\text{ENC}}$ represents the encoder's weights, and $M$ is the dimensionality of the individual latent representation. The encoder architecture is task-condition-specific and designed for condition $A$.

### Decoder

The decoder receives the individual latent representation $z$ and generates the task solver's weights as

$$\Theta_{\mathrm{TS}} = \mathrm{DEC}(z; \Theta_{\mathrm{DEC}}), \tag{3}$$

where $\Theta_{\mathrm{DEC}}$ represents the decoder's weights. Since the decoder determines the task solver's weights, it functions as a hypernetwork (*Ha et al., 2016*; *Karaletsos et al., 2018*).

### Training objective

Although conditions $A$ and $B$ differ, an individual's decision-making system remains consistent across task conditions. We model this using the individual latent representation $z$, linking it to the task solver via the encoder and decoder. For training, we use a behavioral dataset $\{\mathcal{A}_n, \mathcal{B}_n\}_{n \in \mathcal{P}}$ from an individual pool $\mathcal{P}$.

Let $\alpha$ be an action sequence representing individual $n$'s behavior on the source task condition, that is $(\alpha, \phi) \in \mathcal{A}_n$, $n \in \mathcal{P}$. The individual latent representation is derived by $z = \mathrm{ENC}(\alpha, \phi; \Theta_{\mathrm{ENC}})$. The weights of the task solver are then given by $\Theta_{\mathrm{TS}} = \mathrm{DEC}(z; \Theta_{\mathrm{DEC}})$. Subsequently, the task solver, with the given weights, predicts an action sequence for condition $B$ as $\hat{\beta} = \mathrm{TS}(\psi; \Theta_{\mathrm{TS}})$, where $(\beta, \psi) \in \mathcal{B}_n$. We then measure the prediction error between $\hat{\beta}$ and $\beta$ as:

$$L_p(\alpha, \phi, \beta, \psi, \Theta_{\mathrm{ENC}}, \Theta_{\mathrm{DEC}}) = O(\beta, \hat{\beta}), \tag{4}$$

where $\beta$ is an action sequence in $\mathcal{B}_n$ recorded along with the problem $\psi$, and $O(\cdot, \cdot)$ is a suitable loss function (e.g. likelihood-based loss for probabilistic outputs). Using the datasets containing the

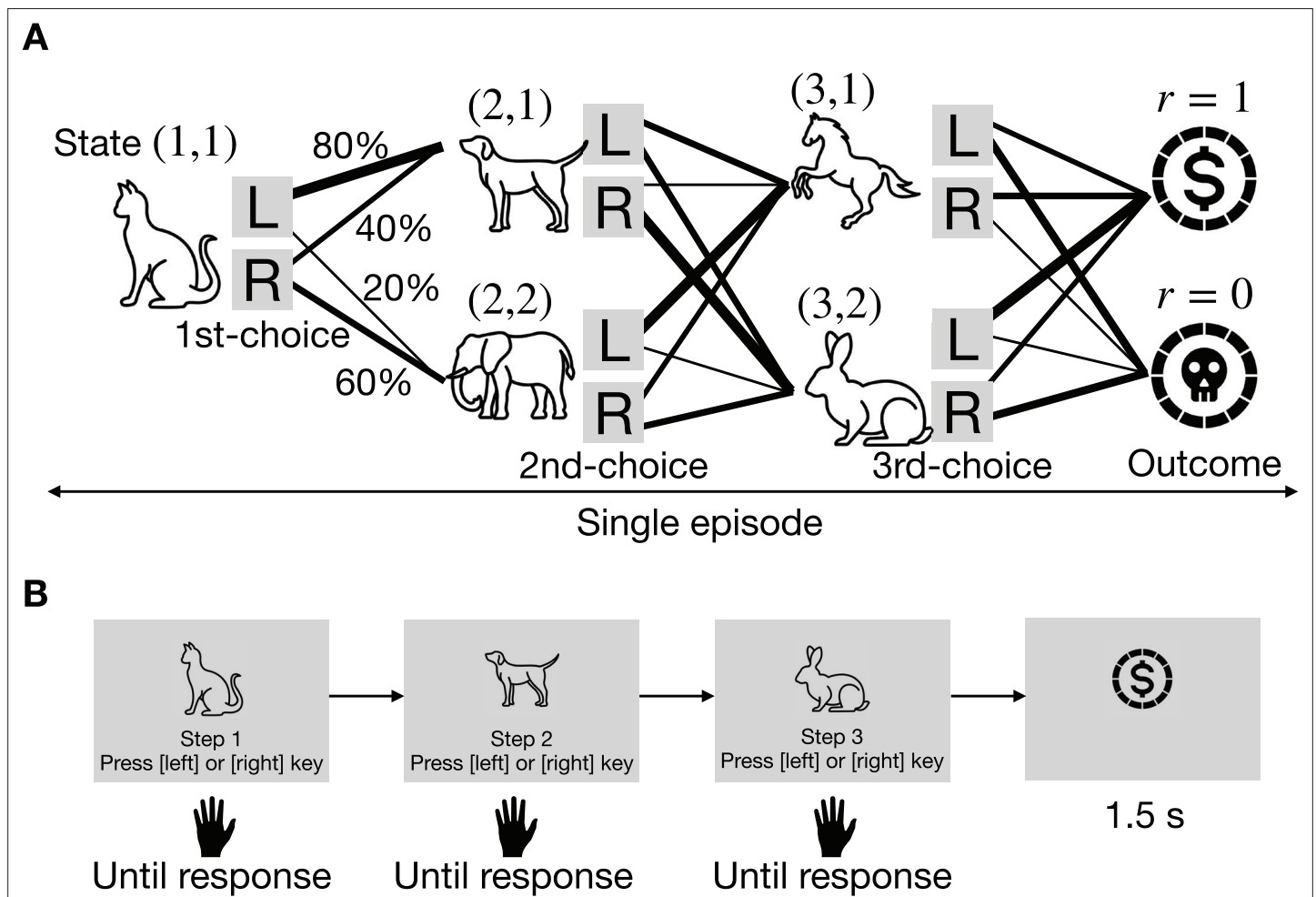

**Figure 12.** The 3-step MDP task. (**A**) Tree diagram illustrating state-action transitions. (**B**) Flow of a single episode in the behavioral experiment for human participants.

behavior of the individual pool $\mathcal{P}$, the weights of the encoder and decoders, $\Theta_{\mathrm{ENC}}$ and $\Theta_{\mathrm{DEC}}$, are optimized by minimizing the total loss:

$$L(\Theta_{\mathrm{ENC}}, \Theta_{\mathrm{DEC}}) = \frac{1}{|\mathcal{P}|} \sum_{n \in \mathcal{P}} \frac{1}{|\mathcal{A}_n|} \sum_{(\alpha, \phi) \in \mathcal{A}_n} \frac{1}{|\mathcal{B}_n|} \sum_{(\beta, \psi) \in \mathcal{B}_n} L_p(\alpha, \phi, \beta, \psi, \Theta_{\mathrm{ENC}}, \Theta_{\mathrm{DEC}}). \tag{5}$$

This section provides a general formulation of individuality transfer across two task conditions. For specific details on task architectures and loss functions, see Experiment on MDP task and Experiment on MNIST task.

## Experiment on MDP task

We validated our individuality transfer framework using two different decision-making tasks: the MDP task and the MNIST task. This section focuses on the MDP tasks, a dynamic multi-step decision-making task.

### Task

At the beginning of each episode, an initial state-cue is presented to the participant. For human participants, the state cue is represented by animal images (*Figure 12*). For the cognitive model (Q-learning agent) and neural network-based model, the state-cue is represented numerically (e.g. (2, 1) for the first task state in the second choice). The participant makes a binary decision (denoted as action $C_1$ or $C_2$) for each step. In the human experiment, these actions correspond to pressing the left or right cursor key. With a certain probability (either 0.8/0.2 or 0.6/0.4), known as the state-action transition probability, the participant transitions to one of two subsequent task states. This process repeats two times for the 2-step MDP and three times in the 3-step MDP. After the final step, the participant receives an outcome: either a reward ($r = 1$) or no reward ($r = 0$). For human participants, rewards were displayed as symbols, as shown in *Figure 12*. Each sequence from initial state-cue presentation to reward delivery constitutes an *episode*.

The state-action transition probability $T(s, a, s')$ from a task state $s$ to a preceding state $s'$ given an action $a$ varies gradually across episodes. With probability $p_{\mathrm{trans}}$, one of the transition probabilities switches to a new set chosen from {(0.8, 0.2), (0.2, 0.8), (0.6, 0.4), (0.4, 0.6)}. Consequently, participants must adjust their decision-making strategy in response to these shifts in transition probabilities to maintain reward maximization.

### Behavioral data collection

We recruited 123 participants via Prolific. All participants provided their informed consent online. This study was approved by the Committee for Human Research at the Graduate School of Engineering, The University of Osaka (Approval number: 5-4-1), and complied with the Declaration of Helsinki. Participants received a base compensation of £4 for completing the entire experiment. A performance-based bonus (£0 to £2, average: £1) was awarded based on rewards earned in the MDP task.

Each participant completed 3 sequences for each step condition (2-step and 3-step MDP tasks), with each sequence comprising 50 episodes. The order of the 2-step and 3-step MDP tasks was randomized across sequences. State-cue assignment (animal images) was randomly determined for each sequence. Participants took a mandatory break (≥1 min) between sequences.

To ensure data quality, we applied exclusion criteria based on average reward, action bias, and response time. Thresholds for these metrics were systematically determined using the interquartile range method on statistics from the initial dataset. Participants were removed from the analysis entirely if their data from any single block fell outside these established ranges. This procedure led to the exclusion of one participant for low average reward (below 0.387 for the 2-step MDP and 0.382 for the 3-step MDP), 23 participants for excessive action bias (outside the 26.3–73.3% range), and 18 for outlier response times (outside the 0.260–1.983 s range). In total, 42 participants (approximately 34%) were excluded, resulting in a final sample of 81 participants for analysis.

### Cognitive model

To model decision-making in the MDP task, we employed a Q-learning agent (*Sutton and Barto, 1998*). At each step $t$, the agent was presented with the current task state $s_t$ and selected an action

$a_t$. The agent maintained $Q$-values, denoted as $Q(s, a)$, for all state-action pairs, where $s$ was a state of the set of all possible task states $\mathcal{S}$ and $a$ was an action of the set of available actions in that state $\mathcal{C}_s$. The probability of selecting action $a$ was determined by a softmax policy:

$$\pi(a) = \frac{\exp(q_{it} Q(s_t, a))}{\sum_{a' \in \mathcal{C}_{s_t}} \exp(q_{it} Q(s_t, a'))}, \tag{6}$$

where $q_{it} > 0$ was a parameter called the inverse temperature or reward sensitivity, controlling the balance between exploration and exploitation.

After selecting action $a_t$, the agent received an outcome $r_t \in \{0, 1\}$ and transitioned to a new state $s_{t+1}$. The $Q$-value for the selected action was updated by

$$Q(s_t, a_t) \leftarrow (1 - q_{lr}) Q(s_t, a_t) + q_{lr}(r_t + q_{dr} \max_{a \in \mathcal{C}_{s_{t+1}}} Q(s_{t+1}, a)), \tag{7}$$

where $q_{lr} \in (0, 1)$ was the learning rate, determining how much newly acquired information replaced existing knowledge, and $q_{dr} \in (0, 1)$ was the discount rate, governing the extent to which future rewards influenced current decision. The $Q$-values are initialized as $q_{init}$ before an agent starts the first episode.

## EIDT model

This section describes the specific models used for individuality transfer in the MDP task.

### Data representation

Since MDP tasks involve sequential decision-making, each action sequence consists of multiple actions within a single session. In our experiment, each participant completed $L$ trials per session, with $L = 100$ for the 2-step MDP and $L = 150$ for the 3-step MDP. The action sequence is represented as $[(s_1, a_1, r_1), \ldots, (s_L, a_L, r_L)]$, where, $s_t$ denotes the task state at trial $t$, $a_t \in C$ represents the action selected from the set $C \equiv \{C_k\}_{k=1}^{K}$ (with $K = 2$ in our task), and $r_t \in \{0, 1\}$ indicates whether a reward was received. In the $M$-step MDP described in **Figure 12**, each task state is represented as $(m, c_m)$, where $m$ denotes the current step within the episode ($m \in \{1, \ldots, M\}$) and $c_m$ corresponds to the cue presented to the participant. The action sequence, denoted as $\alpha$ or $\beta$, consists of a sequence of selected actions $(a_1, \ldots, a_L)$, while a problem, denoted as $\phi$ or $\psi$, is represented as $((s_1, \ldots, s_L), (r_1, \ldots, r_L))$.

### Task solver

Before describing the encoder and decoder, we define the architecture of the task solver, which generates actions for the $M$-step MDP task. The task solver is implemented using a gated recurrent unit (GRU) (**Cho et al., 2014**) with $Q$ cells, where $Q = 4$ for the 2-step task and $Q = 8$ for the 3-step task. At time-step $t$, the GRU takes as input the previous hidden state $\boldsymbol{h}_{t-1} \in \mathbb{R}^Q$, the previous task state $s_{t-1}$, the previous action $a_{t-1}$, the previous reward $r_{t-1}$, and the current task state $s_t$. It then updates the hidden state as

$$\boldsymbol{h}_t = \mathrm{GRU}(s_{t-1}, a_{t-1}, r_{t-1}, s_t, \boldsymbol{h}_{t-1}; \Phi), \tag{8}$$

where $\Phi$ represents the GRU's weights. The updated hidden state is then used to predict the probability of selecting each action through a fully-connected feed-forward layer:

$$\boldsymbol{v}_t = \boldsymbol{W} \boldsymbol{h}_t, \tag{9}$$

where $\boldsymbol{v}_t$ represents the logit scores for each action (unnormalized probabilities), and $\boldsymbol{W} \in \mathbb{R}^{K \times Q}$ is the weight matrix. The probabilities of each action are computed using a softmax layer:

$$\pi(a_t = C_k) = \frac{e^{[\boldsymbol{v}_t]_k}}{\sum_{k'=1,\ldots,K} e^{[\boldsymbol{v}_t]_{k'}}}, \tag{10}$$

where $\pi(a_t = C_k)$ represents the probability of selecting action $C_k$ at time $t$, and $[\boldsymbol{v}_t]_i$ denotes the $i$-th element of $\boldsymbol{v}_t$.

For input encoding, we used a 1-of-K scheme. The step of the MDP task is encoded as $[1, 0, 0]$ for step 1, $[0, 1, 0]$ for step 2, and $[0, 0, 1]$ for step 3. Each task state $s_m$ is represented as $[1, 0]$ or $[0,$

1] to distinguish the two state cues at each step. The participant's action is encoded as $C_1 : [1, 0]$ or $C_2 : [0, 1]$, while the reward is represented as 0: [1, 0] or 1: [0, 1]. These encodings are concatenated to form input sequences.

The task solver $\text{TS}(\psi; \Theta_{\text{TS}})$ generates a sequence of predicted action probabilities $\{(\pi(a_t = C_1), \ldots, \pi(a_t = C_K))\}_{t=1}^{L}$, using the GRU, the fully-connected layer $\boldsymbol{W}$, and the softmax layer. The problem $\psi$ defines the MDP environment, specifying state transitions and reward outcomes in response to selected action.

To evaluate prediction accuracy, the loss function $O(\beta, \hat{\beta})$, defined in **Equation 4**, compares human-performed action $\{\beta, \psi\}$ with those predicted by the task solver, $\{\hat{\beta}, \psi\}$. Notably, the problem $\psi$ is not executed with the task solver; instead, the task solver predicts action probabilities based on the same task state and reward history as in the human behavioral data.

## Encoder and decoder

The encoder $\text{ENC}(\alpha, \phi; \Theta_{\text{ENC}})$ extracts an individual latent representation $z$ from a sequence of actions $\alpha$ corresponding to a given environment $\phi$. The first module of the encoder is a GRU, similar to the task solver, with $R = 32$ cells. The final hidden state $\boldsymbol{h}_L \in \mathbb{R}^R$ serves as the basis for computing the individual latent representation $z \in \mathbb{R}^M$ using a fully-connected feed-forward network with four layers $d(\cdot)$ as $z = d(\boldsymbol{h}_L)$.

The decoder takes the individual latent representation $z$ as input and generates the weights for the task solver by $\Theta_{\text{TS}} = \text{DEC}(z; \Theta_{\text{DEC}})$. The decoder is implemented as a single-layer linear network.

## Experiment on MNIST task

This section describes the specific models used for individuality transfer in handwritten digit recognition (MNIST) task.

### Task

The dataset used in this experiment was originally collected and published by **Rafiei et al., 2024**. In this task, participants were presented with a stimulus image depicting a handwritten digit and were required to respond by pressing the corresponding number key, as illustrated in **Figure 1**. For further details regarding the task design and data collection, refer to **Rafiei et al., 2024**.

### EIDT model
#### Data representation

An action sequence, denoted as $\alpha$ or $\beta$, consists of a single action $a$ and its corresponding response time $b$. The associated problem, represented as $\phi$ or $\psi$, corresponds to a stimulus image. The action $a$ is selected from a set $\{C_1, \ldots C_K\}$. Since the task involves recognizing digits ranging from 0 to 9, the number of possible actions is $K = 10$. The stimulus image, $\phi$ or $\psi$, is an image of size $H \times W$. In this experiment, we adopted the same resolution as (**Rafiei et al., 2024**), setting $H = W = 227$.

#### Task solver

The task solver for the handwritten digit recognition task is based on the model proposed by **Rafiei et al., 2024**. Their model consists of a CNN and an evidence accumulation module. However, since their model represents average human behavior and does not account for individuality differences, we replace the accumulation module with a GRU (**Cheng et al., 2024**) to capture individuality. The CNN module processes the input image and produces an evidence vector $\boldsymbol{e} = \text{CNN}(\psi)$, where $\boldsymbol{e} \in \mathbb{R}^K$ and $\text{CNN}(\cdot)$ is based on the AlexNet architecture (**Krizhevsky et al., 2012**). The weights of the CNN are sampled from a Bayesian neural network (BNN), introducing stochasticity in the output. This stochasticity enables the models to generate human-like, probabilistic decisions.

The stimulus image is fed into the CNN $\boldsymbol{S}$ times, generating $\boldsymbol{S}$ evidence distributions $\boldsymbol{e}_t \in \mathbb{R}^K$ at each time step $t = 0, \ldots, S - 1$. In this study, we set $S = 16$ to match the maximum response time, as described later. Since the CNN weights are stochastically sampled from the BNN, the CNN's output varies even when the same image is input multiple times. To model individuality in decision-making, we introduce a GRU with $Q$ cells ($Q = 4$ in our setup). The GRU receives as input the previous hidden state $\boldsymbol{h}_{t-1} \in \mathbb{R}^Q$ and the current evidence $\boldsymbol{e}_t$, updating its hidden state as

$$\boldsymbol{h}_t = \mathrm{GRU}(\boldsymbol{e}_t, \boldsymbol{h}_{t-1}; \Phi), \qquad (11)$$

where $\Phi$ represents the GRU's network weights. The updated hidden state is passed through a dense layer (as defined in *Equation 9*) and a softmax layer (as defined in *Equation 10*) to generate the probability distribution over possible digit classifications $[P_t(C_1), \ldots, P_t(C_K)]$ at each time step $t$.

To evaluate the prediction error, we compare the action sequences generated by human participants $\{\beta, \psi\}$ with those predicted by the task solver $\{\hat{\beta}, \psi\}$, incorporating response times into these analyses. The actual response time $b$ is converted into an integer time step $\tilde{t}$ using the formula: $\tilde{t} = \mathrm{round}(10b)$. For example, a response time of $b = 0.765$ s is converted to $\tilde{t} = 8$. The likelihood of observed decision is then calculated as $P_{\tilde{t}}(a)$, where $a$ is the actual digit chosen by the participant.

In this task solver, the CNN (driven by BNN) models a visual processing system, while the RNN represents the decision-making system. We assume that the visual system (implemented by CNN and BNN) is shared across all individuals, whereas the decision-making system (implemented by RNN) captures individual differences. Based on this assumption, the CNN and BNN are pretrained using the MNIST dataset (*Krizhevsky et al., 2012*), and their weight distributions are fixed across individuals. The pretraining procedure followed the original methodology (*Rafiei et al., 2024*).

## Encoder and decoder

Since each action sequence contains only a single action, it does not form a true 'sequence'. This makes it challenging to extract individuality from a single data point. To address this, the encoder takes a set of single action sequences as input rather than a single sequence. Specifically, the encoder $\mathrm{ENC}(\{\alpha_u, \phi_u\}_{u=1}^U; \Theta_{\mathrm{ENC}})$ extracts the individual latent representation $z$ from $\boldsymbol{U}$ sets of stimulus images $\phi_u$ and their corresponding responses $\alpha_u$, where $u = 1, \ldots, U$. Here, $\phi_u$ represent the stimulus presented in the $u$-th trial, and $\alpha_u$ represents the corresponding response. The number of action sequences $\boldsymbol{U}$ corresponds to the number of samples available for each individual in the dataset. Since the outputs for these action sequences are just averaged, $\boldsymbol{U}$ can be adjusted flexibly.

The encoder architecture consists of a single CNN module, a single GRU, and a fully connected feed-forward network. The CNN module is identical to the one used in the task solver. Given an input $\phi_u$, let $\boldsymbol{e}_{t,u}$ represent the evidence output from the CNN at time step $t$. The GRU, which consists of $\boldsymbol{R}$ cells ($R = 16$ in our setup), updates its hidden state based on the previous state, the current CNN evidence, and an encoding of the response action by

$$\boldsymbol{h}_{t,u} = \mathrm{GRU}(\boldsymbol{e}_{t,u}, k(a, t, \tilde{t}), \boldsymbol{h}_{t-1,u}; \Psi), \qquad (12)$$

where $\Psi$ represents the network weights. The function $k(a, t, \tilde{t})$ outputs the one-hot encoded action $a$ if $t = \tilde{t}$, and zeros otherwise. The value $\tilde{t}$ represents the converted response time, obtained from the original response time $\boldsymbol{b}$ in the action sequence $\alpha_u$. After processing all $\boldsymbol{U}$ sequences, the final hidden states are averaged across sequences: $\boldsymbol{h} = \frac{1}{U} \sum_{u=1}^{U} \boldsymbol{h}_{L,u}$. The individual latent representation is then computed as $z = d(\boldsymbol{h})$, where $d(\cdot)$ represents a single-layer fully-connected feed-forward network. The decoder, implemented as a single linear layer, takes the individual latent representation $z$ as input and outputs the weights for the task solver.

## Acknowledgements

This work was supported in part by the Japan Society for the Promotion of Science (JSPS) KAKENHI, grant number 22H05163 and 24K15047, and Japan Science and Technology Agency (JST) Advanced International Collaborative Research Program (AdCORP), grant number JPMJKB2307. We appreciate Kaede Hashiguchi and Yuichi Tanaka, Graduate School of Engineering, The University of Osaka, who gave useful comments for this research.

## Additional information

### Funding

| Funder | Grant reference number | Author |
| --- | --- | --- |
| Japan Society for the Promotion of Science | 22H05163 | Hiroshi Higashi |
| Japan Science and Technology Agency | 10.52926/jpmjkb2307 | Hiroshi Higashi |
| Japan Society for the Promotion of Science | 24K15047 | Hiroshi Higashi |

The funders had no role in study design, data collection and interpretation, or the decision to submit the work for publication.

### Author contributions

Hiroshi Higashi, Conceptualization, Data curation, Software, Formal analysis, Funding acquisition, Investigation, Visualization, Methodology, Writing – original draft, Project administration, Writing – review and editing

### Author ORCIDs

Hiroshi Higashi ⬥ https://orcid.org/0000-0001-8880-3411

### Ethics

All participants provided their informed consent online. This study was approved by the Committee for Human Research at the Graduate School of Engineering, The University of Osaka (Approval number: 5-4-1), and compiled with the Declaration of Helsinki.

Reviewer #1 (Public review): https://doi.org/10.7554/eLife.107163.3.sa1
Reviewer #2 (Public review): https://doi.org/10.7554/eLife.107163.3.sa2
Reviewer #3 (Public review): https://doi.org/10.7554/eLife.107163.3.sa3
Author response https://doi.org/10.7554/eLife.107163.3.sa4

## Additional files

### Supplementary files
MDAR checklist

### Data availability

The behavioural data for the MDP task has been made publicly available at https://github.com/hgshrs/indiv_trans (copy archived at *Higashi, 2026*). All code and trained models have been made publicly available at https://github.com/hgshrs/indiv_trans (copy archived at *Higashi, 2026*).

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

# Appendix 1

## Additional results on MDP task

### Parameter fitting of Q-learning for human behaviors

*Appendix 1—figure 1* shows the parameters for the Q-learning model estimated from the behavioral data of human participants. The parameters were estimated separately for the 2-step and 3-step tasks. While the learning rate ($q_{lr}$) and inverse temperature ($q_{it}$) were distributed across a range of values, the discount rate ($q_{dr}$) and the initial Q-value $q_{init}$ were concentrated near 1 and 0, respectively.

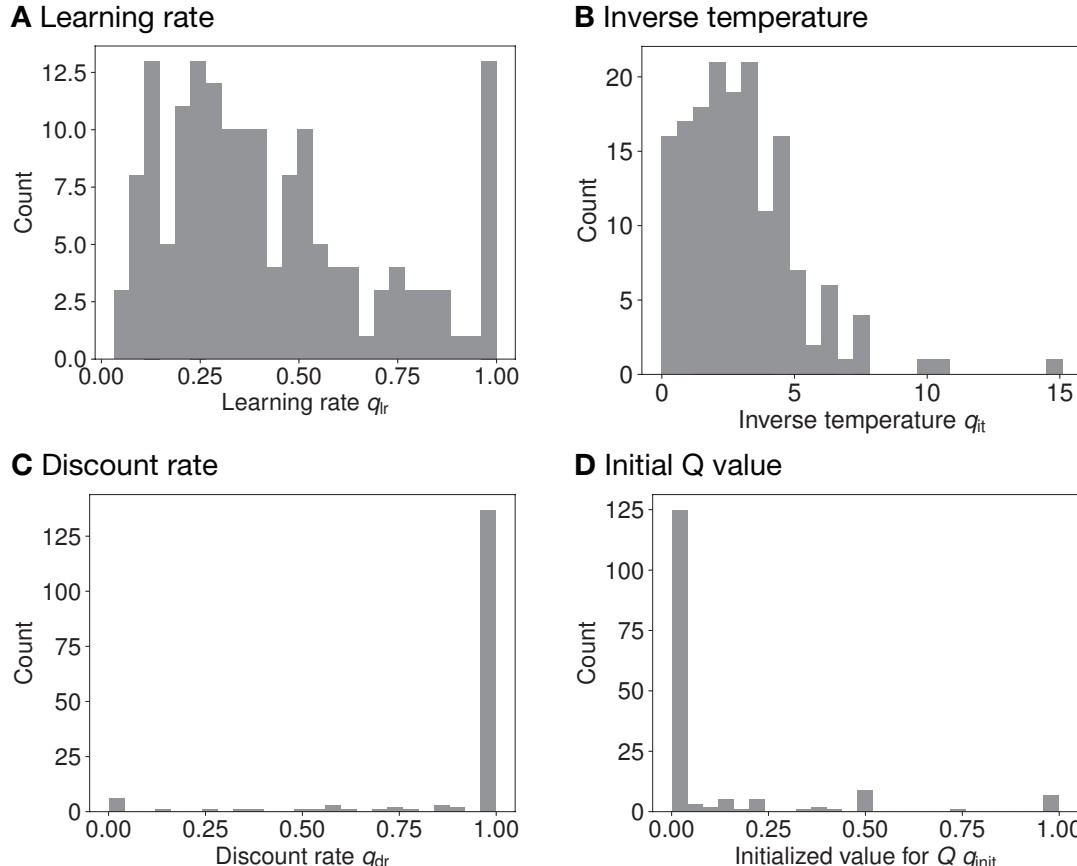

**Appendix 1—figure 1.** Histograms of Q-learning parameters estimated from human participants' behaviors in the MDP tasks. The distributions for the learning rate (**A**) and inverse temperature (**B**) show considerable inter-individual variability, whereas the discount rate (**C**) and initial Q-value (**D**) are relatively consistent across participants.

### EIDT training

The losses for the training and validation samples during the EIDT network training are shown in *Appendix 1—figure 2*. Since we adopted a leave-one-participant-out cross-validation, which results in many training runs, the displayed curves are representatives from a training run that used all participants as training data to illustrate the general convergence pattern. Training was stopped when the validation loss reached its minimum.

**A** 3-step to 2-step transfer

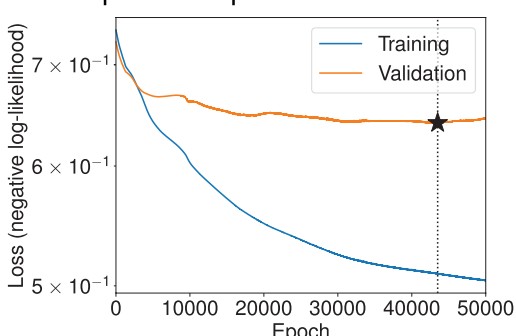

**B** 2-step to 3-step transfer

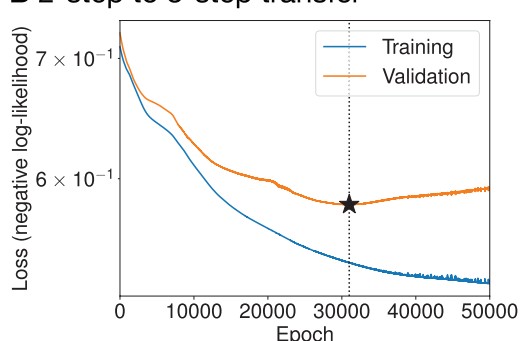

**Appendix 1—figure 2.** Representative training and validation curves for the EIDT model in the MDP task. The plots show the negative log-likelihood loss over training epochs for (**A**) 3-step to 2-step transfer and (**B**) 2-step to 3-step transfer. The star marker indicates the point of early stopping, where the validation loss was minimal.

## Individual latent representation

*Appendix 1—figure 3* visualizes the individual latent representations computed from the behaviors of human participants (squares) and simulated Q-learning agents (dots). Because a different encoder was trained for each fold of the leave-one-participant-out cross-validation, the displayed representations are from a model trained on all participants' data for illustrative purposes.

**A** 3-step to 2-step transfer

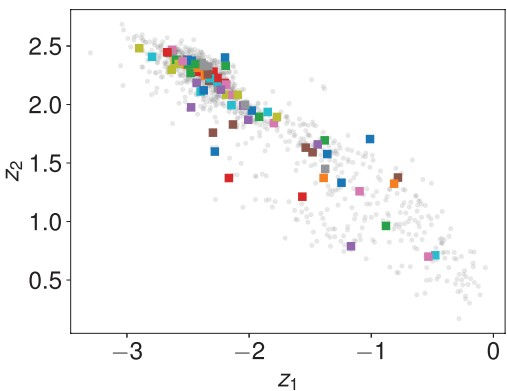

**B** 2-step to 3-step transfer

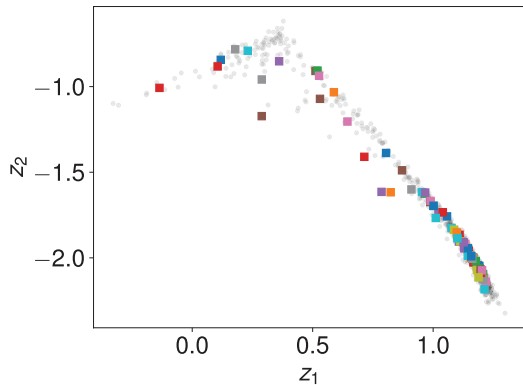

**Appendix 1—figure 3.** Individual latent representations for the MDP task. The plots show the two-dimensional latent space derived from behaviors in (**A**) the 3-step task and (**B**) the 2-step task. Square markers represent human participants, and dot markers represent simulated Q-learning agents.

## Analysis of individual latent representation with a cognitive model

To interpret the latent space, we analyzed the relationship between the Q-learning parameters of simulated agents and their corresponding individual latent representations. We fitted each dimension of the latent representation ($z_i$) using a generalized linear model (GLM) with the agents' learning rate ($q_{lr}$) and inverse temperature ($q_{it}$) as predictors:

$$z_i \sim \text{Normal}\left(\beta_0 + \beta_1 \log(q_{rl}) + \beta_2 \log(q_{it}) + \beta_3 \log(q_{lr}) \log(q_{it})\right), \tag{13}$$

The fitted coefficients are summarized in *Appendix 1—table 1*, and the mapping for the 2-step MDP task is visualized in *Appendix 1—figure 4*. This analysis complements *Appendix 1—figure 6* in the main text, which shows the same mapping for the 3-step task.

**Appendix 1—table 1.** GLM fitting coefficients for the relationship between Q-learning parameters and the individual latent representation.
An asterisk (*) denotes statistical significance ($p < 0.05$).

| | | Coefficients | | | |
|---|---|---|---|---|---|
| Source | Variable | $\beta_0$(bias) | $\beta_1$ ($q_{lr}$) | $\beta_2$ ($q_{it}$) | $\beta_3$ ($q_{lr} \times q_{it}$) |
| 2-step | $z_1$ | −0.613* | 0.096* | 0.064* | −0.035* |
| | $z_2$ | 0.634* | −0.079* | −0.066* | 0.028 |
| 3-step | $z_1$ | 1.157* | −0.388* | −0.121* | −0.148* |
| | $z_2$ | −0.646* | 0.183* | 0.061* | −0.071* |

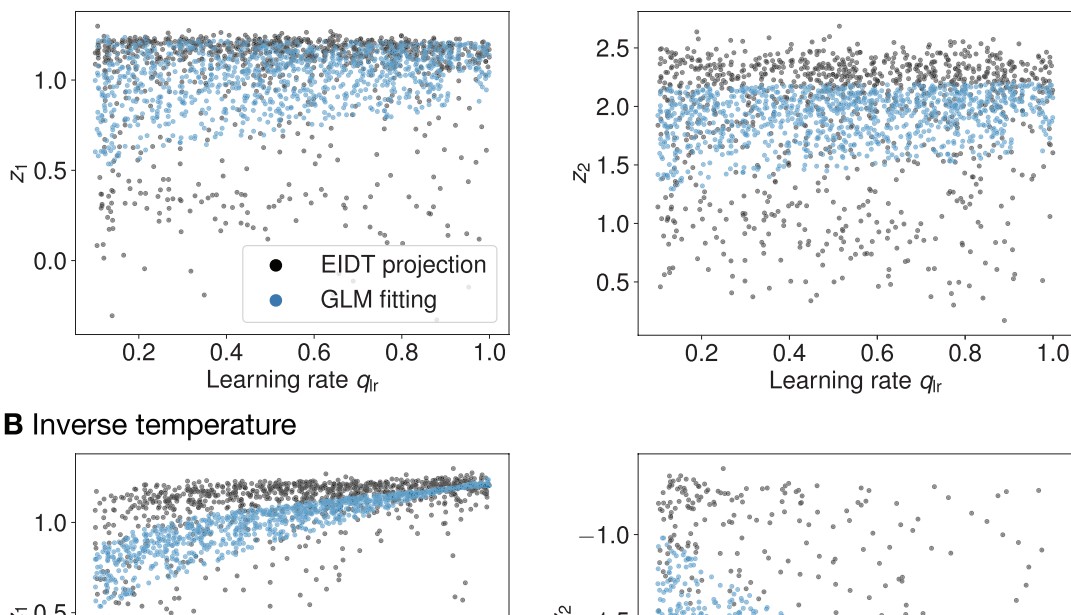

**A** Learning rate

**B** Inverse temperature

**Appendix 1—figure 4.** Mapping of Q-learning parameters to the individual latent space for the 2-step MDP task. Each plot shows one dimension of the latent representation ($z_1$ or $z_2$) as a function of either the learning rate ($q_{lr}$, left) or the inverse temperature ($q_{it}$, right) of simulated Q-learning agents. Black dots represent the latent representation from the agent's behavior, while blue dots show the GLM fit.

## Relationship between prediction performance and latent space for Q-learning agents

We evaluated how the individual latent representation influenced prediction performance for the simulated Q-learning agents. Similar to the analysis on human data, *Appendix 1—figure 5* illustrates the prediction performance as a function of the distance in the individual latent representation space in a cross-individual scenario. Using a GLM, we found that the distance was a significant predictor of both negative log-likelihood (transfer direction 3→2: $\beta_d = 0.094$, $p < 0.001$, 2→3: $\beta_d = 0.164$, $p < 0.001$) and the rate for behavior matched (3→2: $\beta_d = -0.069$, $p < 0.001$, 2→3: $\beta_d = -0.097$, $p < 0.001$). This result shows that, as with human data, prediction performance for an agent degrades as the latent

distance to the source agent increases, confirming that the latent space captures the behavioral tendencies of the simulated agents.

**A** 3-step to 2-step transfer

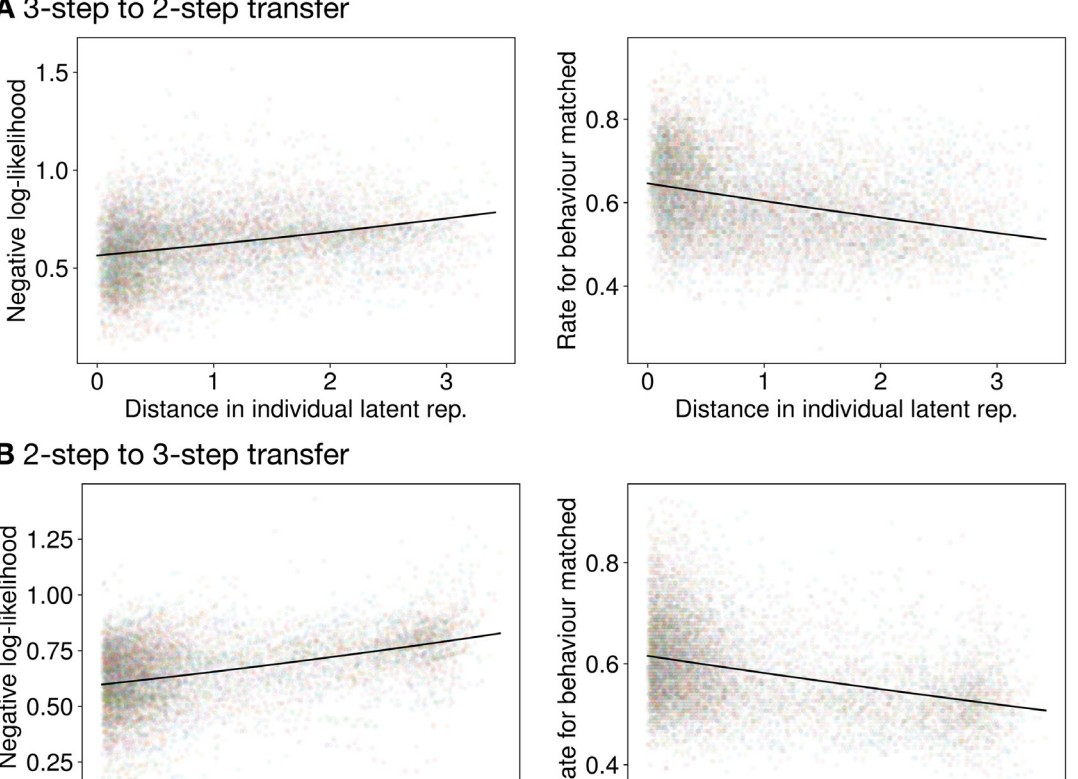

**B** 2-step to 3-step transfer

**Appendix 1—figure 5.** Prediction performances for Q-learning agents as a function of latent space distance. The plots show negative log-likelihood (left) and rate for behavior matched (right) in a cross-individual scenario. (**A**) 3-step to 2-step transfer. (**B**) 2-step to 3-step transfer.

## Additional results on the MNIST task
### EIDT training
*Appendix 1—figure 6* shows representative training and validation loss curves for the EIDT models in the MNIST task. As with the MDP task, these curves are from a model trained from all participants' data for illustrative purposes, showing the typical convergence behavior.

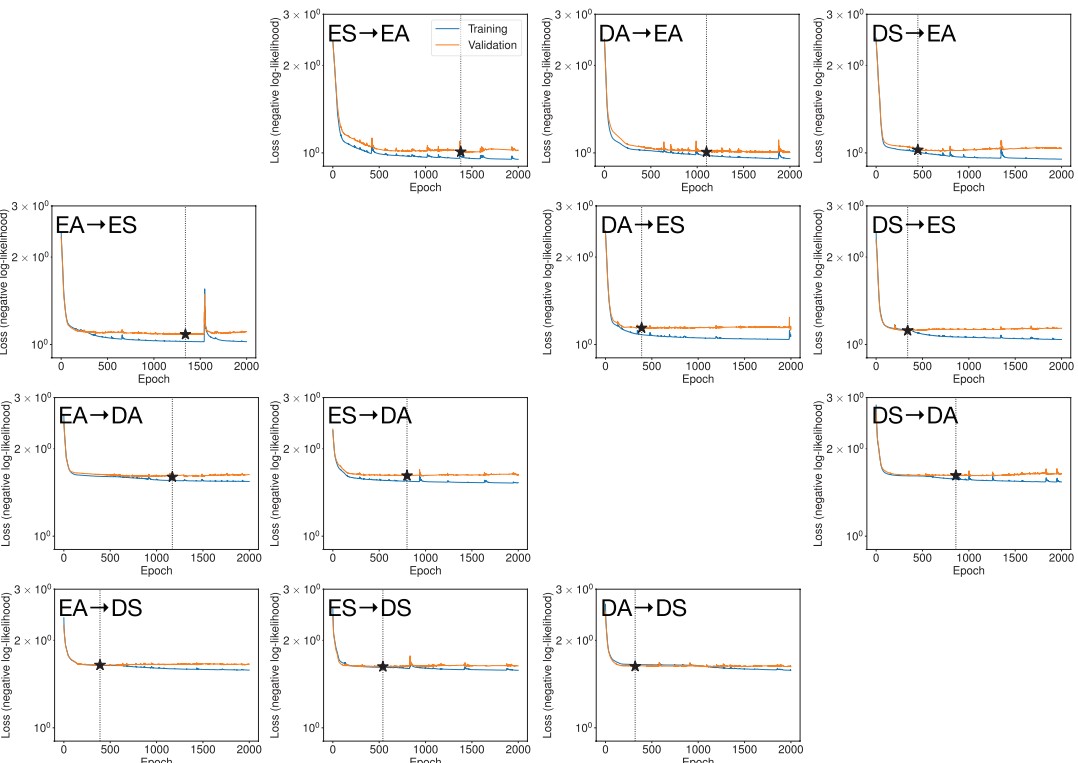

**Appendix 1—figure 6.** Representatives training and validation curves for EIDT models in the MNIST task for each of the 12 transfer directions. Training was stopped at the epoch with the minimum validation loss, indicated by the start marker.

## Individual latent representation

*Appendix 1—figure 7* shows the individual latent representations computed from participants' behaviors for each of the 12 transfer directions in the MNIST task. As we adopted a leave-one-participant-out cross-validation, there were several encoders for each training run. The displayed representations are representatives from a training run using all participants' data.

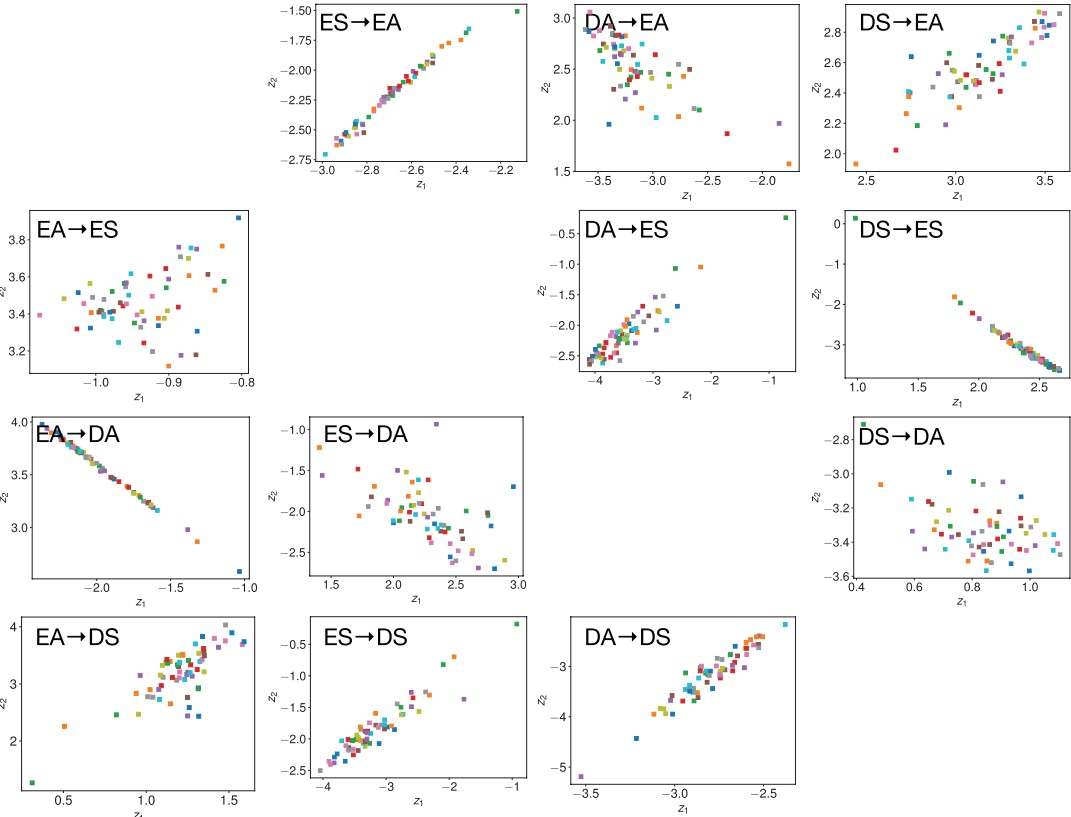

**Appendix 1—figure 7.** Individual latent representations derived from human participants' behaviors in the MNIST task. Each panel shows the two-dimensional latent space generated when using a different experimental condition as the source.

## Relationship between prediction performance and individual latent representation

We performed a cross-individual analysis for the MNIST task, identical to the one conducted for the MDP task. The prediction performance of a task solver derived from one participant (Participant $l$) was evaluated on the data of another participant (Participant $k$), and this performance was analyzed as a function of the distance between their latent representations ($d_{k,l}$).

The results are shown in *Appendix 1—figure 8* (for negative log-likelihood) and *Appendix 1—figure 9* (for rate for behavior matched). In all 12 transfer directions, prediction performance degraded significantly as the distance in the latent space increased. This was confirmed by fitting a GLM:

$$y_{k,l} \sim \text{Gamma}\left(\log(\beta_{\text{participant}_k} + \beta_d d_{k,l} + \beta_0)\right) \tag{14}$$

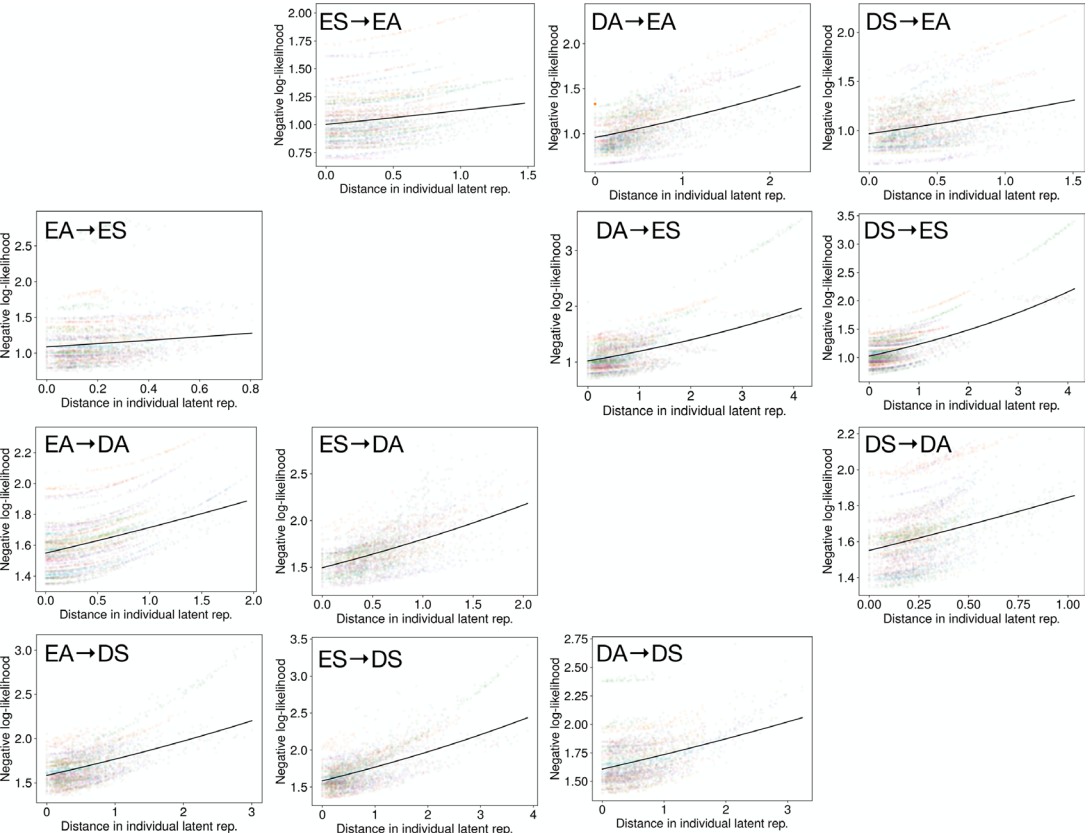

**Appendix 1—figure 8.** Prediction performance (negative log-likelihood) as a function of latent space distance in the MNIST task. Each panel shows the results for one of the 12 transfer directions. The negative log-likelihood (vertical axis) increases as the distance between the source and target individuals' latent representations (horizontal axis) increases, indicating worse prediction performance. The solid line is the GLM fit.

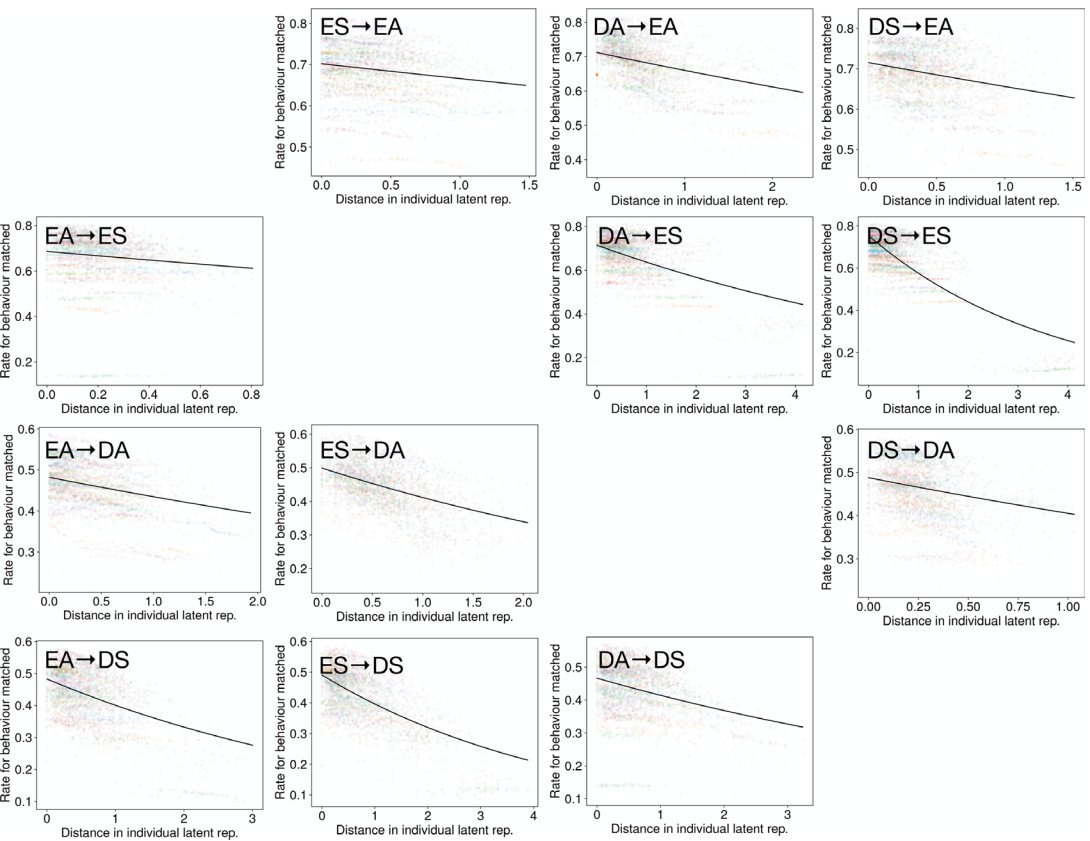

**Appendix 1—figure 9.** Prediction performance (rate for behavior matched) as a function of latent space distance in the MNIST task. Each panel shows the results for one of the 12 transfer directions. The rate for behavior matched (vertical axis) decreases as the distance between individuals' latent representations (horizontal axis) increases. The solid line is the GLM fit.

The fitted coefficients for the distance term ($\beta_d$), shown in **Appendix 1—table 2**, were significant for all transfer directions ($p < 0.001$), reinforcing the conclusion that the latent space captures meaningful individual differences.

**Appendix 1—table 2.** GLM fitting coefficients ($\beta_d$) for the effect of latent space distance on prediction performance in the MNIST task.
All coefficients are statistically significant ($p < 0.001$).

**Negative log-likelihood**

|        |     | Target |       |       |       |
|--------|-----|--------|-------|-------|-------|
|        |     | EA     | ES    | DA    | DS    |
|        | EA  | —      | 0.202 | 0.102 | 0.110 |
|        | ES  | 0.117  | —     | 0.185 | 0.111 |
|        | DA  | 0.199  | 0.158 | —     | 0.076 |
| Source | DS  | 0.201  | 0.186 | 0.174 | —     |

**Rate for behavior matched**

|        |     | Target |       |       |       |
|--------|-----|--------|-------|-------|-------|
|        |     | EA     | ES    | DA    | DS    |

*Continued on next page*

*Continued*

**Rate for behavior matched**

|        |     |        |        |        |        |
|--------|-----|--------|--------|--------|--------|
|        | EA  | —      | −0.142 | −0.103 | −0.187 |
|        | ES  | −0.053 | —      | −0.193 | −0.214 |
|        | DA  | −0.076 | −0.115 | —      | −0.118 |
| Source | DS  | −0.086 | −0.270 | −0.185 | —      |

