## [Editor Report · eLife Assessment]

This revised paper provides a **valuable** and novel neural network-based framework for parameterizing individual differences and predicting individual decision-making across task conditions. The methods and analyses are **solid** yet could benefit from further validation of the superiority of the proposed framework against other baseline models. With these concerns addressed, this study would offer a proof-of-concept neural network approach to scientists working on the generalization of cognitive skills across contexts.

---

## [Referee Report · Reviewer #1 (Public review)]

Summary

The manuscript presents EIDT, a framework that extracts an "individuality index" from a source task to predict a participant's behaviour in a related target task under different conditions. However, the evidence that it truly enables cross-task individuality transfer is not convincing.

Strengths

The EIDT framework is clearly explained, and the experimental design and results are generally well-described. The performance of the proposed method is tested on two distinct paradigms: a Markov Decision Process (MDP) task (comparing 2-step and 3-step versions) and a handwritten digit recognition (MNIST) task under various conditions of difficulty and speed pressure. The results indicate that the EIDT framework generally achieved lower prediction error compared to baseline models and that it was better at predicting a specific individual's behaviour when using their own individuality index compared to using indices from others.

Furthermore, the individuality index appeared to form distinct clusters for different individuals, and the framework was better at predicting a specific individual's behaviour when using their own derived index compared to using indices from other individuals.

Comments on revisions:

I thank the author for the additional analyses. They have fully addressed all of my previous concerns, and I have no further recommendations.

---

## [Referee Report · Reviewer #2 (Public review)]

This paper introduces a framework for modeling individual differences in decision-making by learning a low-dimensional representation (the "individuality index") from one task and using it to predict behaviour in a different task. The approach is evaluated on two types of tasks: a sequential value-based decision-making task and a perceptual decision task (MNIST). The model shows improved prediction accuracy when incorporating this learned representation compared to baseline models.

The motivation is solid, and the modelling approach is interesting, especially the use of individual embeddings to enable cross-task generalization. That said, several aspects of the evaluation and analysis could be strengthened.

(1) The MNIST SX baseline appears weak. RTNet isn't directly comparable in structure or training. A stronger baseline would involve training the GRU directly on the task without using the individuality index-e.g., by fixing the decoder head. This would provide a clearer picture of what the index contributes.

(2) Although the focus is on prediction, the framework could offer more insight into how behaviour in one task generalizes to another. For example, simulating predicted behaviours while varying the individuality index might help reveal what behavioural traits it encodes.

(3) It's not clear whether the model can reproduce human behaviour when acting on-policy. Simulating behaviour using the trained task solver and comparing it with actual participant data would help assess how well the model captures individual decision tendencies.

(4) Figures 3 and S1 aim to show that individuality indices from the same participant are closer together than those from different participants. However, this isn't fully convincing from the visualizations alone. Including a quantitative presentation would help support the claim.

(5) The transfer scenarios are often between very similar task conditions (e.g., different versions of MNIST or two-step vs three-step MDP). This limits the strength of the generalization claims. In particular, the effects in the MNIST experiment appear relatively modest, and the transfer is between experimental conditions within the same perceptual task. To better support the idea of generalizing behavioural traits across tasks, it would be valuable to include transfers across more structurally distinct tasks.

(6) For both experiments, it would help to show basic summaries of participants' behavioural performance. For example, in the MDP task, first-stage choice proportions based on transition types are commonly reported. These kinds of benchmarks provide useful context.

(7) For the MDP task, consider reporting the number or proportion of correct choices in addition to negative log-likelihood. This would make the results more interpretable.

(8) In Figure 5, what is the difference between the "% correct" and "% match to behaviour"? If so, it would help to clarify the distinction in the text or figure captions.

(9) For the cognitive model, it would be useful to report the fitted parameters (e.g., learning rate, inverse temperature) per individual. This can offer insight into what kinds of behavioural variability the individuality index might be capturing.

(10) A few of the terms and labels in the paper could be made more intuitive. For example, the name "individuality index" might give the impression of a scalar value rather than a latent vector, and the labels "SX" and "SY" are somewhat arbitrary. You might consider whether clearer or more descriptive alternatives would help readers follow the paper more easily.

(11) Please consider including training and validation curves for your models. These would help readers assess convergence, overfitting, and general training stability, especially given the complexity of the encoder-decoder architecture.

Comments on revisions:

Thank you to the authors for the updated manuscript. The authors have addressed the majority of my concerns, and the paper is now in a much better form.

Regarding my previous Comment 6, I still believe it would be helpful to include a graph similar to what is typically reported for these tasks-specifically, a breakdown of choices based on rare versus common transitions (see Model-Based Influences on Humans' Choices and Striatal Prediction Errors, Figure 2). Presenting this for both the actual behaviour and the simulated data would strengthen the paper and allow for clearer comparison.

---

## [Referee Report · Reviewer #3 (Public review)]

Summary:

This work presents a novel neural network-based framework for parameterizing individual differences in human behavior. Using two distinct decision-making experiments, the author demonstrates the approach's potential and claims it can predict individual behavior (1) within the same task, (2) across different tasks, and (3) across individuals. While the goal of capturing individual variability is compelling and the potential applications are promising, the claims are weakly supported, and I find that the underlying problem is conceptually ill-defined.

Strengths:

The idea of using neural networks for parameterizing individual differences in human behavior is novel, and the potential applications can be impactful.

Weaknesses:

(1) To demonstrate the effectiveness of the approach, the authors compare a Q-learning cognitive model (for the MDP task) and RTNet (for the MNIST task) against the proposed framework. However, as I understand it, neither the cognitive model nor RTNet is designed to fit or account for individual variability. If that is the case, it is unclear why these models serve as appropriate baselines. Isn't it expected that a model explicitly fitted to individual data would outperform models that do not? If so, does the observed superiority of the proposed framework simply reflect the unsurprising benefit of fitting individual variability? I think the authors should either clarify why these models constitute fair control or validate the proposed approach against stronger and more appropriate baselines.

(2) It's not very clear in the results section what it means by having a shorter within-individual distance than between-individual distances. Related to the comment above, is there any control analysis performed for this? Also, this analysis appears to have nothing to do with predicting individual behavior. Is this evidence toward successfully parameterizing individual differences? Could this be task-dependent, especially since the transfer is evaluated on exceedingly similar tasks in both experiments? I think a bit more discussion of the motivation and implications of these results will help the reader in making sense of this analysis.

(3) The authors have to better define what exactly he meant by transferring across different "tasks" and testing the framework in "more distinctive tasks". All presented evidence, taken at face value, demonstrated transferring across different "conditions" of the same task within the same experiment. It is unclear to me how generalizable the framework will be when applied to different tasks.

(4) Conceptually, it is also unclear to me how plausible it is that the framework could generalize across tasks spanning multiple cognitive domains (if that's what is meant by more distinctive). For instance, how can an individual's task performance on a Posner task predict task performance on the Cambridge face memory test? Which part of the framework could have enabled such a cross-domain prediction of task performance? I think these have to be at least discussed to some extent, since without it the future direction is meaningless.

(5) How is the negative log-likelihood, which seems to be the main metric for comparison, computed? Is this based on trial-by-trial response prediction or probability of responses, as what usually performed in cognitive modelling?

(6) None of the presented evidence is cross-validated. The authors should consider performing K-fold cross-validation on the train, test, and evaluation split of subjects to ensure robustness of the findings.

(7) The authors excluded 25 subjects (20% of the data) for different reasons. This is a substantial proportion, especially by the standards of what is typically observed in behavioral experiments. The authors should provide a clear justification for these exclusion criteria and, if possible, cite relevant studies that support the use of such stringent thresholds.

(8) The authors should do a better job of creating the figures and writing the figure captions. It is unclear which specific claim the authors are addressing with the figure. For example, what is the key message of Figure 2C regarding transfer within and across participants? Why are the stats presentation different between the Cognitive model and the EIDT framework plots? In Figure 3, it's unclear what these dots and clusters represent and how they support the authors' claim that the same individual forms clusters. And isn't this experiment have 98 subjects after exclusion, this plot has way less than 98 dots as far as I can tell. Furthermore, I find Figure 5 particularly confusing, as the underlying claim it is meant to illustrate is unclear. Clearer figures and more informative captions are needed to guide the reader effectively.

(9) I also find the writing somewhat difficult to follow. The subheadings are confusing, and it's often unclear which specific claim the authors are addressing. The presentation of results feels disorganized, making it hard to trace the evidence supporting each claim. Also, the excessive use of acronyms (e.g., SX, SY, CG, EA, ES, DA, DS) makes the text harder to parse. I recommend restructuring the results section to be clearer and significantly reducing the use of unnecessary acronyms.

Comments on revisions:

The authors have addressed my previous comments with great care and detail. I appreciate the additional analyses and edits. I have no further comments.

---

## [Author Response]

The following is the authors’ response to the original reviews.

**Public Reviews:**

**Reviewer #1 (Public review):**
Because the "source" and "target" tasks are merely parameter variations of the same paradigm, it is unclear whether EIDT achieves true crosstask transfer. The manuscript provides no measure of how consistent each participant's behaviour is across these variants (e.g., two- vs threestep MDP; easy vs difficult MNIST). Without this measure, the transfer results are hard to interpret. In fact, Figure 5 shows a notable drop in accuracy when transferring between the easy and difficult MNIST conditions, compared to transfers between accuracy-focused and speedfocused conditions. Does this discrepancy simply reflect larger withinparticipant behavioural differences between the easy and difficult settings? A direct analysis of intra-individual similarity for each task pair and how that similarity is related to EIDT's transfer performance is needed.

Thank you for your insightful comment. We agree that the tasks used in our study are variations of the same paradigm. Accordingly, we have revised the manuscript to consistently frame our findings as demonstrating individuality transfer "across task conditions" rather than "across distinct tasks."

In response to your suggestion, we have conducted a new analysis to directly investigate the relationship between individual behavioural patterns and transfer performance. As show in the new Figures 4, 11, S8, and S9, we found a clear relationship between the distance in the space of individual latent representation (called individuality index in the previous manuscript) and prediction performance. Specifically, prediction accuracy for a given individual's behaviour degrades as the latent representation of the model's source individual becomes more distant. This result directly demonstrates that our framework captures meaningful individual differences that are predictive of transfer performance across conditions.

We have also expanded the Discussion (Lines 332--343) to address the potential for applying this framework to more structurally distinct tasks, hypothesizing that this would rely on shared underlying cognitive functions.

Related to the previous comment, the individuality index is central to the framework, yet remains hard to interpret. It shows much greater within-participant variability in the MNIST experiment (Figure S1) than in the MDP experiment (Figure 3). Is such a difference meaningful? It is hard to know whether it reflects noisier data, greater behavioural flexibility, or limitations of the model.

Thank you for raising this important point about interpretability. To enhance the interpretability of the individual latent representation, we have added a new analysis for the MDP task (see Figures 6 and S4). By applying our trained encoder to data from simulated Q-learning agents with known parameters, we demonstrate that the dimensions of the latent space systematically map onto the agents' underlying cognitive parameters (learning rate and inverse temperature). This analysis provides a clearer interpretation by linking our model's data-driven representation to established theoretical constructs.

Regarding the greater within-participant variability observed in the MNIST task (visualized now in Figure S7), this could be attributed to several factors, such as greater behavioural flexibility in the perceptual task. However, disentangling these potential factors is complex and falls outside the primary scope of the current study, which prioritizes demonstrating robust prediction accuracy across different task conditions.

The authors suggests that the model's ability to generalize to new participants "likely relies on the fact that individuality indices form clusters and individuals similar to new participants exist in the training participant pool". It would be helpful to directly test this hypothesis by quantifying the similarity (or distance) of each test participant's individuality index to the individuals or identified clusters within the training set, and assessing whether greater similarity (or closer proximity) to the clusters in the training set is associated with higher prediction accuracy for those individuals in the test set.

Thank you for this excellent suggestion. We have performed the analysis you proposed to directly test this hypothesis. Our new results, presented in Figures 4, 11, S5, S8, and S9, quantify the distance between the latent representation of a test participant and that of the source participant used to generate the prediction model.

The results show a significant negative correlation: prediction accuracy consistently decreases as the distance in the latent space increases. This confirms that generalization performance is directly tied to the similarity of behavioural patterns as captured by our latent representation, strongly supporting our hypothesis.

**Reviewer #2 (Public review):**
The MNIST SX baseline appears weak. RTNet isn't directly comparable in structure or training. A stronger baseline would involve training the GRU directly on the task without using the individuality index-e.g., by fixing the decoder head. This would provide a clearer picture of what the index contributes.

We agree that a more direct baseline is crucial for evaluating the contribution of our transfer mechanism. For the Within-Condition Prediction scenario, the comparison with RTNet was intended only to validate that our task solver architecture could achieve average humanlevel task performance (Figure 7).

For the critical Cross-Condition Transfer scenario, we have now implemented a stronger and more appropriate baseline, which we call ``task solver (source).'' This model has the same architecture as our EIDT task solver but is trained directly on the source task data of the specific test participant. As shown in revised Figure 9, our EIDT framework significantly outperforms this direct-training approach, clearly demonstrating the benefit of the individuality transfer mechanism.

Although the focus is on prediction, the framework could offer more insight into how behaviour in one task generalizes to another. For example, simulating predicted behaviours while varying the individuality index might help reveal what behavioural traits it encodes.

Thank you for this valuable suggestion. To provide more insight into the encoded behavioural traits, we have conducted a new analysis linking the individual latent representation to a theoretical cognitive model. As detailed in the revised manuscript (Figures 6 and S4), we applied our encoder to simulated data from Q-learning agents with varying parameters. The results show a systematic relationship between the latent space coordinates and the agents' learning rates and inverse temperatures, providing a clearer interpretation of what the representation captures.

It's not clear whether the model can reproduce human behaviour when acting on-policy. Simulating behaviour using the trained task solver and comparing it with actual participant data would help assess how well the model captures individual decision tendencies.

We have added the suggested on-policy evaluation (Lines 195--207). In the revised manuscript (Figure 5), we present results from simulations where the trained task solvers performed the MDP task. We compared their performance (total reward and rate of the highly-rewarding action selected) against their corresponding human participants. The strong correlations observed demonstrate that our model successfully captures and reproduces individual-specific behavioural tendencies in an onpolicy setting.

Figures 3 and S1 aim to show that individuality indices from the same participant are closer together than those from different participants. However, this isn't fully convincing from the visualizations alone. Including a quantitative presentation would help support the claim.

We agree that the original visualizations of inter- and intraparticipant distances was not sufficiently convincing. We have therefore removed that analysis. In its place, we have introduced a more direct and quantitative analysis that explicitly links the individual latent representation to prediction performance (see Figures 4, 11, S5, S8, and S9). This new analysis demonstrates that prediction error for an individual is a function of distance in the latent space, providing stronger evidence that the representation captures meaningful, individual-specific information.

The transfer scenarios are often between very similar task conditions (e.g., different versions of MNIST or two-step vs three-step MDP). This limits the strength of the generalization claims. In particular, the effects in the MNIST experiment appear relatively modest, and the transfer is between experimental conditions within the same perceptual task. To better support the idea of generalizing behavioural traits across tasks, it would be valuable to include transfers across more structurally distinct tasks.

We agree with this limitation and have revised the manuscript to be more precise. We now frame our contribution as "individuality transfer across task conditions" rather than "across tasks" to accurately reflect the scope of our experiments. We have also expanded the Discussion section (Line 332-343) to address the potential and challenges of applying this framework to more structurally distinct tasks, noting that it would likely depend on shared underlying cognitive functions.

For both experiments, it would help to show basic summaries of participants' behavioural performance. For example, in the MDP task, first-stage choice proportions based on transition types are commonly reported. These kinds of benchmarks provide useful context.

We have added behavioral performance summaries as requested. For the MDP task, Figure 5 now compares the total reward and rate of highlyrewarding action selected between humans and our model. For the MNIST task, Figure 7 shows the rate of correct responses for humans, RTNet, and our task solver across all conditions. These additions provide better context for the model's performance.

For the MDP task, consider reporting the number or proportion of correct choices in addition to negative log-likelihood. This would make the results more interpretable.

Thank you for the suggestion. To make the results more interpretable, we have added a new prediction performance metric: the rate for behaviour matched. This metric measures the proportion of trials where the model's predicted action matches the human's actual choice. This is now included alongside the negative log-likelihood in Figures 2, 3, 4, 8, 9, and 11.

In Figure 5, what is the difference between the "% correct" and "% match to behaviour"? If so, it would help to clarify the distinction in the text or figure captions.

We have clarified these terms in the revised manuscript. As defined in the Result section (Lines 116--122, 231), "%correct" (now "rate of correct responses") is a measure of task performance, whereas "%match to behaviour" (now "rate for behaviour matched") is a measure of prediction accuracy.

For the cognitive model, it would be useful to report the fitted parameters (e.g., learning rate, inverse temperature) per individual. This can offer insight into what kinds of behavioural variability the individual latent representation might be capturing.

We have added histograms of the fitted Q-learning parameters for the human participants in Supplementary Materials (Figure S1). This analysis revealed which parameters varied most across the population and directly informed the design of our subsequent simulation study with Q-learning agents (see response to Comment 2-2), where we linked these parameters to the individual latent representation (Lines 208--223).

A few of the terms and labels in the paper could be made more intuitive. For example, the name "individuality index" might give the impression of a scalar value rather than a latent vector, and the labels "SX" and "SY" are somewhat arbitrary. You might consider whether clearer or more descriptive alternatives would help readers follow the paper more easily.

We have adopted the suggested changes for clarity.

"Individuality index" has been changed to "individual latent representation".

"Situation SX" and "Situation SY" have been renamed to the more descriptive "Within-Condition Prediction" and "Cross-Condition Transfer", respectively.

We have also added a table in Figure 7 to clarify the MNIST condition acronyms (EA/ES/DA/DS).

Please consider including training and validation curves for your models. These would help readers assess convergence, overfitting, and general training stability, especially given the complexity of the encoder-decoder architecture.

Training and validation curves for both the MDP and MNIST tasks have been added to Supplementary Materials (Figure S2 and S6) to show model convergence and stability.

**Reviewer #3 (Public review):**
To demonstrate the effectiveness of the approach, the authors compare a Q-learning cognitive model (for the MDP task) and RTNet (for the MNIST task) against the proposed framework. However, as I understand it, neither the cognitive model nor RTNet is designed to fit or account for individual variability. If that is the case, it is unclear why these models serve as appropriate baselines. Isn't it expected that a model explicitly fitted to individual data would outperform models that do not? If so, does the observed superiority of the proposed framework simply reflect the unsurprising benefit of fitting individual variability? I think the authors should either clarify why these models constitute fair control or validate the proposed approach against stronger and more appropriate baselines.

Thank you for raising this critical point. We wish to clarify the nature of our baselines:

For the MDP task, the cognitive model baseline was indeed designed to account for individual variability. We estimated its parameters (e.g., learning rate) from each individual's source task behaviour and then used those specific parameters to predict their behaviour in the target task. This makes it a direct, parameter-based transfer model and thus a fair and appropriate baseline for individuality transfer.

For the MNIST task, we agree that the RTNet baseline was insufficient for evaluating individual-level transfer in the "Cross-Condition Transfer" scenario. We have now introduced a much stronger baseline, the "task solver (source)," which is trained specifically on the source task data of each test participant. Our results (Figure 9) show that the EIDT framework significantly outperforms this more appropriate, individualized baseline, highlighting the value of our transfer method over direct, within-condition fitting.

It's not very clear in the results section what it means by having a shorter within-individual distance than between-individual distances. Related to the comment above, is there any control analysis performed for this? Also, this analysis appears to have nothing to do with predicting individual behavior. Is this evidence toward successfully parameterizing individual differences? Could this be task-dependent, especially since the transfer is evaluated on exceedingly similar tasks in both experiments? I think a bit more discussion of the motivation and implications of these results will help the reader in making sense of this analysis.

We agree that the previous analysis on inter- and intra-participant distances was not sufficiently clear or directly linked to the model's predictive power. We have removed this analysis from the manuscript. In its place, we have introduced a new, more direct analysis (Figures 4, 11, S5, S8, and S9) that demonstrates a quantitative relationship between the distance in the latent space and prediction accuracy. This new analysis shows that prediction error for an individual increases as a function of this distance, providing much stronger and clearer evidence that our framework successfully parameterizes meaningful individual differences.

The authors have to better define what exactly he meant by transferring across different "tasks" and testing the framework in "more distinctive tasks". All presented evidence, taken at face value, demonstrated transferring across different "conditions" of the same task within the same experiment. It is unclear to me how generalizable the framework will be when applied to different tasks.Conceptually, it is also unclear to me how plausible it is that the framework could generalize across tasks spanning multiple cognitive domains (if that's what is meant by more distinctive). For instance, how can an individual's task performance on a Posner task predict task performance on the Cambridge face memory test? Which part of the framework could have enabled such a cross-domain prediction of task performance? I think these have to be at least discussed to some extent, since without it the future direction is meaningless.

We agree with your assessment and have corrected our terminology throughout the manuscript. We now consistently refer to the transfer as being "across task conditions" to accurately describe the scope of our findings.

We have also expanded our Discussion (Line 332-343) to address the important conceptual point about cross-domain transfer. We hypothesize that such transfer would be possible if the tasks, even if structurally different, rely on partially shared underlying cognitive functions (e.g., working memory). In such a scenario, the individual latent representation would capture an individual's specific characteristics related to that shared function, enabling transfer. Conversely, we state that transfer between tasks with no shared cognitive basis would not be expected to succeed with our current framework.

How is the negative log-likelihood, which seems to be the main metric for comparison, computed? Is this based on trial-by-trial response prediction or probability of responses, as what usually performed in cognitive modelling?

The negative log-likelihood is computed on a trial-by-trial basis. It is based on the probability the model assigned to the specific action that the human participant actually took on that trial. This calculation is applied consistently across all models (cognitive models, RTNet, and EIDT). We have added sentences to the Results section to clarify this point (Lines 116--122).

None of the presented evidence is cross-validated. The authors should consider performing K-fold cross-validation on the train, test, and evaluation split of subjects to ensure robustness of the findings.

All prediction performance results reported in the revised manuscript are now based on a rigorous leave-one-participant-out cross-validation procedure to ensure the robustness of our findings. We have updated the

Methods section to reflect this (Lines 127--129 and 229).

For some purely illustrative visualizations (e.g., plotting the entire latent space in Figures S3 and S7), we used a model trained on all participants' data to provide a single, representative example and avoid clutter. We have explicitly noted this in the relevant figure captions.

The authors excluded 25 subjects (20% of the data) for different reasons. This is a substantial proportion, especially by the standards of what is typically observed in behavioral experiments. The authors should provide a clear justification for these exclusion criteria and, if possible, cite relevant studies that support the use of such stringent thresholds.

We acknowledge the concern regarding the exclusion rate. The previous criteria were indeed empirical. We have now implemented more systematic exclusion procedure based on the interquartile range of performance metrics, which is detailed in Section 4.2.2 (Lines 489--498). This revised, objective criterion resulted in the exclusion of 42 participants (34% of the initial sample). While this rate is high, we attribute it to the online nature of the data collection, where participant engagement can be more variable. We believe applying these strict criteria was necessary to ensure the quality and reliability of the behavioural data used for modeling.

The authors should do a better job of creating the figures and writing the figure captions. It is unclear which specific claim the authors are addressing with the figure. For example, what is the key message of Figure 2C regarding transfer within and across participants? Why are the stats presentation different between the Cognitive model and the EIDT framework plots? In Figure 3, it's unclear what these dots and clusters represent and how they support the authors' claim that the same individual forms clusters. And isn't this experiment have 98 subjects after exclusion, this plot has way less than 98 dots as far as I can tell. Furthermore, I find Figure 5 particularly confusing, as the underlying claim it is meant to illustrate is unclear. Clearer figures and more informative captions are needed to guide the reader effectively.

We agree that several figures and analyses in the original manuscript were unclear, and we have thoroughly revised our figures and their captions to improve clarity.

The confusing analysis in the old Figures 2C and 5 (Original/Others comparison) have been completely removed. The unclear visualization of the latent space for the test pool (old Figure 3 showing representations only from test participants) has also been removed to avoid confusion. For visualization of the overall latent space, we now use models trained on all data (Figures S3 and S7) and have clarified this in the captions. In place of these removed analyses, we have introduced a new, more intuitive "cross-individual" analysis (presented in Figures 4, 11, S5, S8, and S9). As explained in the new, more detailed captions, this analysis directly plots prediction performance as a function of the distance in latent space, providing a much clearer demonstration of how the latent representation relates to predictive accuracy.

I also find the writing somewhat difficult to follow. The subheadings are confusing, and it's often unclear which specific claim the authors are addressing. The presentation of results feels disorganized, making it hard to trace the evidence supporting each claim. Also, the excessive use of acronyms (e.g., SX, SY, CG, EA, ES, DA, DS) makes the text harder to parse. I recommend restructuring the results section to be clearer and significantly reducing the use of unnecessary acronyms.

Thank you for this feedback. We have made significant revisions to improve the clarity and organization of the manuscript. We have renamed confusing acronyms: "Situation SX" is now "Within- Condition Prediction," and "Situation SY" is now "Cross-Condition Transfer." We also added a table to clarify the MNIST condition acronyms (EA/ES/DA/DS) in Figure 7.

The Results section has been substantially restructured with clearer subheadings.